# An Even More Optimal Stochastic Optimization Algorithm: Minibatching and Interpolation Learning

**Blake Woodworth**
Toyota Technological Institute at Chicago
blake@ttic.edu

**Nathan Srebro**
Toyota Technological Institute at Chicago
nati@ttic.edu

## Abstract

We present and analyze an algorithm for optimizing smooth and convex or strongly convex objectives using minibatch stochastic gradient estimates. The algorithm is optimal with respect to its dependence on both the minibatch size and minimum expected loss simultaneously. This improves over the optimal method of Lan [17], which is insensitive to the minimum expected loss; over the optimistic acceleration of Cotter et al. [10], which has suboptimal dependence on the minibatch size; and over the algorithm of Liu and Belkin [19], which is limited to least squares problems and is also similarly suboptimal with respect to the minibatch size. Applied to interpolation learning, the improvement over Cotter et al. and Liu and Belkin translates to a linear, rather than square-root, parallelization speedup.

## 1 Introduction

The massive scale of many modern machine learning models and datasets give rise to complex, high-dimensional training objectives that can be very computationally expensive to optimize. To reduce the computational cost, it is therefore important to devise optimization algorithms that can leverage parallelism to reduce the amount of time needed to train. Stochastic first-order methods, which use stochastic estimates of the gradient of the training objective, are by far the most common approach and these methods can be directly improved by using minibatch stochastic gradient estimates, which are easy to parallelize across multiple computing cores or devices. Accordingly, we propose and analyze an optimal accelerated minibatch stochastic gradient descent algorithm.

Our analysis exploits that while training machine learning models, it is typically possible to drive the loss either all the way to zero, or at least very close to zero, and the performance of our algorithm improves as the minimum value of the loss approaches zero. This property has multiple names in different contexts. In learning theory, it is common to show fast rates under the assumption of "realizability"—meaning the data can be fit perfectly by the model, i.e. the loss can be driven to zero. Even when the problem is not exactly realizable, it is sometimes possible to derive "optimistic rates" that interpolate between fast rates for realizable learning and the slower agnostic rates [33]. In the context of "interpolation learning," there has recently been great interest in understanding training "overparametrized" models—which have many more parameters than there are training examples, generally meaning that many settings of the parameters would attain zero training loss—both in terms of optimization [2–4, 9, 15] and generalization [6, 35, 38]. Finally, in the optimization literature, there have been efforts to prove optimistic rates depending on the minimum value of the objective or on the variance of the stochastic gradients a the minimizer [10, 19–21, 23, 31]. Regardless of the name, these ideas are all based on the same fundamental concept of exploiting the fact that the minimum value of the objective is nearly zero.

35th Conference on Neural Information Processing Systems (NeurIPS 2021).

**Our contributions**

In Section 3, we present and analyze an accelerated minibatch SGD algorithm for optimizing smooth and convex objectives using minibatch stochastic gradient estimates. Our method closely resembles the methods of Lan [17] and Cotter et al. [10], but with different stepsizes and momentum parameters, and a tighter analysis. Importantly, our algorithm enjoys a linear speedup in the minibatch size all the way up to a critical threshold beyond which larger minibatches do not help. In contrast, Lan and Cotter et al.'s bounds have a worse, sublinear speedup and a correspondingly higher critical threshold.

In Section 4, we show that a modified version of our algorithm can attain substantially faster convergence when the objective satisfies a certain quadratic growth condition, which is a relaxation of strong convexity. As part of our analysis, we simplify and generalize a restarting technique [13, 18, 24, 26, 29, 30], which we show amounts to a reduction *from* strongly convex optimization *to* convex optimization. The reduction in the other direction is a well-known tool in optimization analysis, but our result shows that it goes both ways and it may be of more general interest.

In Sections 5 and 6, we prove that our methods are optimal with respect to both the minibatch size and also the minimum value of the loss in the settings we consider. We then explain how our guarantees demonstrate a linear speedup in the minibatch size, which improves over a sublinear speedup in previous work [10, 19].

Finally, in Section 7, we extend our results to a related setting where the bound on the minimum value of the loss is replaced by a bound on the variance of the stochastic gradients at the optimum [as in, e.g. 8, 14, 21, 23, 31]. Under this condition, we establish the optimal error achievable by any learning rule, including non-first-order methods, and we show that the optimal convergence rate is nevertheless achieved by SGD—a first-order method without acceleration. Further, we show that the accelerated optimization rate, $T^{-2}$, is unattainable using minibatches of size 1, but with larger minibatches, our accelerated minibatch SGD method can match the optimal error of SGD using a substantially smaller parallel runtime but the same number of samples.

## 2 Setting and Background

We consider a generic stochastic optimization problem

$$\min_{w \in \mathbb{R}^d} \{ L(w) := \mathbb{E}_{z \sim \mathcal{D}}[\ell(w; z)] \} \tag{1}$$

This problem captures, for instance, supervised machine learning where $z = (x, y)$ is feature vector and label pair; $\ell(w; (x, y))$ is the loss of a model parametrized by $w$ on that example; and $L(w)$ is the expected loss. We note that there are two possible interpretations of $L$ depending on what $\mathcal{D}$ corresponds to. We can take $\mathcal{D}$ to be the uniform distribution over a set of training examples, in which case $L$ is the training loss. Alternatively, we can take $\mathcal{D}$ to be the population distribution, in which case $L$ is the population risk. An advantage of the latter view is that optimization guarantees directly imply good performance on the population, however, computing an independent, unbiased stochastic gradient estimate requires a fresh sample from the distribution, so only "one-pass" methods are possible. Nevertheless, our algorithm and analysis apply equally well in either viewpoint.

We consider optimizing objectives on $\mathbb{R}^d$, but we are most interested in dimension-free analysis, i.e. one that does not explicitly depend on the dimension, and our algorithm's guarantees hold even in infinite dimension. Indeed, in many applications including machine learning, the dimension can be very large, so an explicit reliance on the dimension would often lead to impractical bounds.

We study the family of optimization algorithms that attempt to minimize (1) using $T$ sequential stochastic gradient estimates with minibatches of size $b$ of the form

$$g(w_t) = \frac{1}{b} \sum_{i=1}^{b} \nabla \ell(w_t; z^i) \quad \text{for i.i.d. } z^1, \dots, z^b \sim \mathcal{D} \tag{2}$$

at points $w_1, \dots, w_T$ of the algorithm's choice. Later, we will argue that our proposed method is optimal with respect to all of the algorithms in this family. Because it is easy to parallelize the computation of the $b$ stochastic gradients in the minibatch, $T$ roughly captures the runtime of the algorithm, while $n = bT$, the total number of samples used, captures its sample complexity.

In our analysis, we will rely on several assumptions about the losses $\ell$ and the expected loss $L$.

**Assumption 1.** *For almost every $z \sim \mathcal{D}$, $\ell(w; z)$ is non-negative, convex, and $H$-smooth w.r.t. $w$, i.e.*

$$\forall_{w,u,z} \; \ell(u; z) + \langle \nabla \ell(u; z), \, w - u \rangle \leq \ell(w; z) \leq \ell(u; z) + \langle \nabla \ell(u; z), \, w - u \rangle + \frac{H}{2} \|w - u\|^2$$

This assumption holds, for instance, for training linear models with a smooth, convex, and non-negative loss function, as in least squares problems or logistic regression.

**Assumption 2.** *The expected loss $L$ is convex and has minimum value $L^* = \min_w L(w)$, which is attained at a point $w^*$ with $\|w^*\| \leq B$.*

The minimum value of the loss, $L^*$, is the key quantity in our analysis, and we will prove an "optimistic rate" for our algorithm, meaning it performs increasingly well as $L^* \to 0$. The key idea is to use $L^*$ to bound the variance of the stochastic gradient estimates at the point $w^*$. Following prior work [10, 32, 33], we observe that under Assumptions 1 and 2, we can upper bound (see Lemma 3)

$$\mathbb{E}\|\nabla \ell(w^*; z) - \nabla L(w^*)\|^2 \leq 2HL^* \tag{3}$$

The $H$-Lipschitzness of $\nabla \ell$ allows us to upper bound the variance of the stochastic gradients at other, non-minimizing points $w$ too. In contrast, a common practice in the optimization literature is to simply assert that the variance of the stochastic gradient estimates is uniformly upper bounded, i.e.

$$\sup_w \mathbb{E}\|\ell(w; z) - \nabla L(w)\|^2 \leq \sigma^2 \tag{4}$$

This bound is very convenient, but it can fail to hold even in very simple cases like least squares regression, where the variance grows with $\|w\|^2$. Therefore, while faster rates may be achieved under the condition (4), this is quite strong and may not correspond well with problems of interest.

For many optimization problems, particularly those arising from training machine learning models, the minimum of the loss, $L^*$, can be expected to be small. For example, machine learning models are often trained in the interpolation regime [2–4, 6, 9, 15, 35, 38], where the training loss is underdetermined, so there are many settings of the parameters which achieve exactly zero training loss. When $\mathcal{D}$ is the empirical distribution (see above), the interpolation regime therefore corresponds to $L^* = 0$. However, even if $\mathcal{D}$ is the population distribution, machine learning problems are often realizable, or nearly realizable, meaning the population risk can also be driven to zero or close to it.

In the optimization literature, it is often possible to show enhanced guarantees under the favorable condition of $\lambda$-strong convexity. However, the assumption of strong convexity is somewhat at odds with our goal of studying the interpolation or near-interpolation setting. For example, the simplest example of interpolation learning is underdetermined least squares, where the number of observations is less than the dimension; in this case, the empirical covariance is degenerate so the problem is not strongly convex. Furthermore, assuming strong convexity *and $L^* \approx 0$ and $\ell$* is non-negative is very strong, and puts very strong constraints on the objective function which may often fail to hold.

We consider instead a relaxation of strong convexity, which only requires the objective to grow faster than the squared distance to the set of minimizers, which need not be a singleton [7, 12, 22]. This condition *does* hold for underdetermined least squares—with $\lambda$ equal to the smallest *non-zero* eigenvalue of the covariance—and for many other problems with non-unique minimizers, and it turns out to be sufficient to achieve the faster rates that typically arise from strong convexity.

**Assumption 3.** *The expected loss $L$ is convex; it has minimum value $L^* = \min_w L(w)$; $L(0) - L^* \leq \Delta$; and $L$ satisfies the following growth condition for all $w$:*

$$L(w) - L^* \geq \frac{\lambda}{2} \min_{w^* \in \arg\min_w L(w)} \|w - w^*\|^2$$

**Related Work**

The foundation of much of the work on optimal stochastic optimization algorithms is the accelerated SGD variant, AC-SA, of Lan [17]. This algorithm is analyzed in a different setting, where $L$ is $H$-smooth, $B$-bounded, and convex and where the stochastic gradients have uniformly bounded variance (4), and under these conditions, it is optimal, with guarantee

$$\mathbb{E}L(\hat{w}) - L^* \leq c \cdot \left( \frac{HB^2}{T^2} + \frac{\sigma B}{\sqrt{bT}} \right) \tag{5}$$

---
**Algorithm 1** Accelerated Minibatch SGD
---
$w_0^{\text{ag}} = w_0 = 0$
**for** $t = 0, 1, \ldots, T - 1$ **do**
$\quad \beta_t = 1 + \frac{t}{6}$ and $\gamma_t = \gamma(t+1)$ for $\gamma = \min\left\{ \frac{1}{12H}, \frac{b}{24H(T+1)}, \sqrt{\frac{bB^2}{HL^*T^3}} \right\}$
$\quad w_t^{\text{md}} = \beta_t^{-1} w_t + (1 - \beta_t^{-1}) w_t^{\text{ag}}$
$\quad \tilde{w}_{t+1} = w_t - \gamma_t g_t(w_t^{\text{md}})$ where $g_t(w_t^{\text{md}}) = \frac{1}{b} \sum_{i=1}^{b} \nabla \ell(w_t^{\text{md}}; z_t^i)$ for i.i.d. $z_t^1, \ldots, z_t^b \sim \mathcal{D}$
$\quad w_{t+1} = \min\left\{ 1, \frac{B}{\|\tilde{w}_{t+1}\|} \right\} \tilde{w}_{t+1}$
$\quad w_{t+1}^{\text{ag}} = \beta_t^{-1} w_{t+1} + (1 - \beta_t^{-1}) w_t^{\text{ag}}$
**Return:** $w_T^{\text{ag}}$
---

In our setting, however, there is no such explicit bound on the gradient variance, and at the same time, Lan's analysis does not exploit the bound on $L^*$ to achieve a better rate. Under our Assumptions 1 and 2, it is possible to derive a variance upper bound $\sigma^2 = 2H^2B^2 + 4HL^*$, which yields

$$\mathbb{E}L(\hat{w}) - L^* \leq c \cdot \left( \frac{HB^2}{T^2} + \frac{HB^2}{\sqrt{bT}} + \sqrt{\frac{HB^2L^*}{bT}} \right) \tag{6}$$

The first and third terms of this bound are tight, but we will later show that this guarantee is suboptimal because the denominator of the second term can be improved to $bT$. When $L^* \approx HB^2$—the largest possible value of $L^*$ under mild conditions[1]—we will later show that this bound is tight, so we can interpret this modification of Lan's guarantee as being optimal only when $L^*$ is large.

In other, more directly comparable existing work, Cotter et al. [10] propose and analyze a minibatch SGD algorithm and an accelerated variant similar to AC-SA under our Assumptions 1 and 2. Their accelerated method is guaranteed to achieve suboptimality

$$\mathbb{E}L(\hat{w}) - L^* \leq c \cdot \left( \frac{HB^2}{T^2} + \frac{HB^2}{\sqrt{bT}} + \frac{HB^2\sqrt{\log T}}{bT} + \sqrt{\frac{HB^2L^*}{bT}} \right) \tag{7}$$

In the special case of $b = 1$, we will show that this is optimal (up to a minor $\sqrt{\log T}$ factor). However, the $\sqrt{bT}$ dependence in the second term can be improved to $bT$, so Cotter et al.'s analysis is suboptimal when $b > 1$. We also extend our analysis to the Assumptions 1 and 3.

In other related work, Liu and Belkin [19] propose a different minibatch accelerated SGD method. In the special case of least squares objectives—where $\ell(w; (x, y)) = \frac{1}{2}(\langle w, x \rangle - y)^2$—that also satisfy Assumptions 1 and 3 with $L^* = 0$, they show that their algorithm attains suboptimality

$$\mathbb{E}L(\hat{w}) - L^* \leq c \cdot \left( \Delta \exp\left( -\frac{c'\sqrt{\lambda}T}{\sqrt{H}} \right) + \Delta \exp\left( -\frac{c'\lambda\sqrt{b}T}{H} \right) \right) \tag{8}$$

As with Cotter et al., we show that the $\sqrt{bT}$ dependence of this guarantee is suboptimal and can be improved to $bT$. In addition, our analysis goes beyond the special case of least squares with $L^* = 0$.

In other related work, Bassily et al. [5] study non-accelerated algorithms in a similar setting. Zhang et al. [40] consider a related setting where the losses $\ell$ are Lipschitz rather than smooth, and where the dimension of the problem is sufficiently small relative to the other problem parameters. Zhang and Zhou [39] study our smooth setting but with an additional restriction that $\ell$ is Lipschitz and $L$ is strongly convex, but show only polynomial convergence versus our linear convergence. Srebro et al. [33] study the performance of the empirical risk minimizer under Assumptions 1 and 2 as well as stochastic first-order methods with $b = 1$; their methods are optimal for $b = 1$, but they do not analyze the effect of the minibatch size. Vaswani et al. [36] consider a different noise assumption, which is similar to requiring $L^* = 0$.

---
[1]The minimum $L^* \leq O(HB^2)$, for example, when $\ell(w; z) = 0$ is realized within a ball of radius $O(B)$.

## 3 A Better Accelerated Minibatch SGD Method

Our Algorithm 1 is very similar to the AC-SA algorithm of Lan [17] and to the AG algorithm of Cotter et al. [10]; the difference is that we use different stepsize and momentum parameters and provide a tighter analysis for our setting. Our method provides the following guarantee:

**Theorem 1.** *Let $\ell$ and $L$ satisfy Assumptions 1 and 2, then Algorithm 1 guarantees for a universal constant $c$*

$$\mathbb{E}L(w_T^{ag}) - L^* \leq c \cdot \left( \frac{HB^2}{T^2} + \frac{HB^2}{bT} + \sqrt{\frac{HB^2L^*}{bT}} \right)$$

We prove this in Appendix A using a similar approach as the analysis for AC-SA of Lan [17]. As discussed previously, Lan's analysis relies on a uniform upper bound on the stochastic gradient variance on the set $\{w : \|w\| \leq B\}$. While an upper bound can be derived in our setting, it resembles $H^2B^2$ which is too large to achieve good performance. Our analysis, in contrast, exploits the fact that the points $w_0^{\text{md}}, \ldots, w_{T-1}^{\text{md}}$ at which the stochastic gradients are actually computed approach $w^*$ as the algorithm proceeds, which implies that the stochastic gradient variance decreases over time. It is difficult to identify precisely why our analysis improves in its dependence on the minibatch size, $b$, compared with Cotter et al.'s bound. However, the primary difference between our analyses is that Cotter et al. use stepsizes $\gamma_t \propto t^p$ for $p < 1$, while our stepsizes scales linearly with $t$. Our choice leads to somewhat simpler computations, which may explain our tighter bound.

## 4 The Reduction and Faster Rates

Our algorithm can also be extended to the setting of Assumptions 1 and 3 using a restarting argument [13, 18, 24, 26, 29, 30]. The proof is based on a simple idea, which we will show amounts to a reduction from strongly convex optimization to convex optimization. However, previous applications of restarting schemes tend to involve relatively complex and specialized proofs. Here, we simplify and generalize the approach, and we present it in a way that will hopefully be convenient for future use. We then apply it to provide an enhanced guarantee for Algorithm 1 under Assumptions 1 and 3.

Convex optimization algorithms guarantee reducing the value of the objective by an amount that depends on some measure of the distance between the initialization and a minimizer of the objective. The key idea in the analysis is that when the objective is $\lambda$-strongly convex, reducing the value of the objective implies reducing the *distance* to the minimizer:

$$\|w - w^*\|^2 \leq \frac{2(L(w) - L^*)}{\lambda} \tag{9}$$

Therefore, if we apply an algorithm for convex objectives to a strongly convex objective several times in succession, each time reducing the suboptimality by a constant factor, then (1) roughly $\log 1/\epsilon$ applications will suffice to reach $\epsilon$-suboptimality and (2) reducing the suboptimality by a constant factor will get no harder with each application since the distance to the optimum is decreasing. Using this idea allows us to take an algorithm with a guarantee for convex objectives and derive an algorithm with a corresponding, better guarantee for strongly convex objectives. We also show that strong convexity can be replaced by a weaker condition that generalizes Assumption 3.

In order to present the reduction, we first define

**Definition 1.** *Given $\psi : \mathbb{R}^d \to \mathbb{R}_+$, we say that $L$ satisfies the $(\lambda, \psi)$-growth condition (hereafter $(\lambda, \psi)$-GC) if for all $w$*

$$L(w) - L^* \geq \lambda\psi(w)$$

For example, the $(\lambda, \psi)$-GC for $\psi(w) = \frac{1}{2}\|w - w^*\|^2$ is equivalent to $\lambda$-strong convexity, and the condition in Assumption 3 is equivalent to the $(\lambda, \psi)$-GC for $\psi(w) = \arg\min_{w^* \in \arg\min_w L(w)} \|w - w^*\|^2$. The second ingredient of the reduction is the "time" needed by an algorithm to optimize convex or strongly convex objectives:

**Definition 2.** *Let $\mathcal{L}$ be any set of convex functions. We define* $\text{Time}(\epsilon, B, \psi, \mathcal{L}, \mathcal{A})$ *to be the time needed by the algorithm $\mathcal{A}$ to find a point $\hat{w}$ with $\mathbb{E}L(\hat{w}) - L^* \leq \epsilon$ given a point $w_0$ with $\mathbb{E}\psi(w_0) \leq B$, for any $L \in \mathcal{L}$. Similarly, we define* $\text{Time}_\lambda(\epsilon, \Delta, \psi, \mathcal{L}, \mathcal{A})$ *to be the time needed by $\mathcal{A}$ to find $\hat{w}$ with $\mathbb{E}L(\hat{w}) - L^* \leq \epsilon$ given a point $w_0$ with $\mathbb{E}L(w_0) - L^* \leq \Delta$, for any $L \in \mathcal{L}$ that also satisfies the $(\lambda, \psi)$-GC.*

**Algorithm 2** GC2Cvx$(\mathcal{A}, \theta)$

---

Given: $w_0$ s.t. $\mathbb{E}L(w_0) - L^* \leq \Delta$
**for** $t = 1, 2, \ldots, T = \lceil \log_\theta \frac{\Delta}{\epsilon} \rceil$ **do**
    Set $w_t$ to be the output of $\mathcal{A}$ initialized a $w_{t-1}$ after $\mathsf{Time}\big(\theta^{-t}\Delta, \theta^{1-t}\frac{\Delta}{\lambda}, \psi, \mathcal{L}, \mathcal{A}\big)$
Return $w_T$

---

We are deliberately vague about the precise meaning of "time" here. Typically, it would correspond to the number of iterations of the algorithm, but it could also count the number of times the algorithm accesses a certain oracle, or even the wall-clock time of an implementation of the algorithm, but it can correspond to essentially any (subadditive) property of the algorithm. With these definitions in hand, we present the reduction, Algorithm 2, which guarantees:

**Theorem 2.** *For any algorithm $\mathcal{A}$ and $\theta > 1$,* GC2Cvx$(\mathcal{A}, \theta)$ *defined in Algorithm 2 guarantees*

$$\mathsf{Time}_\lambda(\epsilon, \Delta, \psi, \mathcal{L}, \mathsf{GC2Cvx}(\mathcal{A}, \theta)) \leq \sum_{t=1}^{\lceil \log_\theta \frac{\Delta}{\epsilon} \rceil} \mathsf{Time}\left( \theta^{-t}\Delta, \theta^{1-t}\frac{\Delta}{\lambda}, \psi, \mathcal{L}, \mathcal{A} \right)$$

We prove this very concisely in Appendix B and also discuss several example applications of the Theorem in order to give a better sense of how it can be applied. This reduction complements a standard tool in the optimization toolbox, which goes in the opposite direction. Specifically, it is very common to take an algorithm for optimizing $\lambda$-strongly convex objectives and to apply it to convex objectives by optimizing the $\lambda$-strongly convex surrogate $L_\lambda(w) = L(w) + \frac{\lambda}{2}\|w\|^2$. Sometimes, this approach is slightly suboptimal, and the similar, multistep procedure of Allen-Zhu and Hazan [1] is required, nevertheless, it was already known that strongly convex optimization is, for this reason, "easier" than convex optimization. Our result shows that the reverse is also true, so strongly convex and convex optimization are, in a certain sense, equally hard.

In combination, these two reductions going in each direction appear to be optimal in a certain way. In all of the examples we have tried, composing both reductions—converting a convex algorithm to a strongly convex one, and then converting that strongly convex algorithm back to a convex one—results in only a constant-factor degradation in the guarantee. We have also consistently observed that applying the reduction, Algorithm 2, to an optimal algorithm for convex optimization yields an optimal algorithm for strongly-convex optimization, and we conjecture that this holds universally.

Before moving on, we note that the reduction Algorithm 2 can result in an unusual method. For example, the algorithm GC2Cvx(Alg 1, $e$) analyzed below involves repeated invocations of Algorithm 1, resulting in the stepsize and momentum parameters being reset periodically. As a whole, it resembles Algorithm 1, however it has a strange, non-monotonic stepsize and momentum schedule. Therefore, while this technique can be useful for deriving new optimization algorithms with better theoretical guarantees, these methods may not always be aethetically pleasing or practical, and it may still be useful to study more "natural" methods that can be directly analyzed under the $(\lambda, \psi)$-GC.

Finally, we apply Theorem 2 to Algorithm 1 and derive the following stronger guarantee under Assumptions 1 and 3, which we prove in Appendix C:

**Theorem 3.** *Let $\ell$ and $L$ satisfy Assumptions 1 and 3, then the output of* GC2Cvx(Alg 1, $e$) *guarantees for universal constants $c, c'$*

$$\mathbb{E}L(\hat{w}) - L^* \leq c \cdot \left( \Delta \exp\left( -\frac{c'\sqrt{\lambda}T}{\sqrt{H}} \right) + \Delta \exp\left( -\frac{c'\lambda bT}{H} \right) + \frac{HL^*}{\lambda bT} \right)$$

## 5 The Optimality of Our Algorithms

Here, we argue that the guarantees in Theorems 1 and 3 are optimal. First, it is well-known that even without any noise, i.e. $\ell(w; z) = L(w)$, any first-order method will have error at least [25]

$$L(\hat{w}) - L^* \geq c \cdot \frac{HB^2}{T^2} \qquad \text{or} \qquad L(\hat{w}) - L^* \geq c \cdot \Delta \exp\left( -c' \cdot \frac{\sqrt{\lambda}T}{\sqrt{H}} \right) \tag{10}$$

under Assumptions 1 and 2 or Assumptions 1 and 3, respectively (in fact, the latter holds even under the stronger condition of $\lambda$-strong convexity). Therefore, the first terms in our method's guarantees in Theorems 1 and 3 are tight, and correspond to the rate achieved by accelerated gradient descent [28].

For the remaining second and third terms in each guarantee, we prove a lower bound that applies for *any* learning rule that uses $n$ i.i.d. samples from the distribution, even non-first-order methods. This lower bound also applies to minibatch first-order algorithms with $n = bT$ being the total number of stochastic gradient estimates, since each $\nabla \ell(w, z)$ is computed using a single sample.

**Theorem 4.** *For $\ell(w; (x, y)) = \frac{1}{2}(\langle w, x \rangle - y)^2$ the square loss, for any learning algorithm that takes $n$ samples as input, there exists a distribution over $(x, y)$ pairs such that $\ell$ and $L$ satisfy Assumptions 1 and 2, and for a universal constant c, the algorithm's output will have error at least*

$$\mathbb{E}L(\hat{w}) - L^* \geq c \cdot \left( \frac{HB^2}{n} + \sqrt{\frac{HB^2 L^*}{n}} \right)$$

*Similarly, there exists a distribution over $(x, y)$ pairs such that $\ell$ and $L$ satisfy Assumptions 1 and 3 (and, in fact, $L$ is $\lambda$-strongly convex), and for a universal constant c, the algorithm's output will have error at least*

$$\mathbb{E}L(\hat{w}) - L^* \geq c \cdot \left( \Delta \cdot \mathbb{1}_{n \leq \frac{H}{2\lambda}} + \min\left\{ \frac{HL^*}{\lambda n}, \Delta \right\} \right)$$

We prove this in Appendix D using an argument similar to that of Srebro et al. [33]. The lower bound for deterministic first-order optimization, (10), plus the sample complexity lower bound, Theorem 4, together imply a lower bound for the minibatch first-order algorithms that we consider, which matches our guarantees:

**Corollary 1.** *There exists $\ell(w; z)$ such that for any algorithm that uses $T$ minibatch stochastic gradients of size $b$, there exists a distribution over $z$ such that $\ell$ and $L$ satisfy Assumptions 1 and 2, and for a universal constant c, the algorithm's output will have error at least*

$$\mathbb{E}L(\hat{w}) - L^* \geq c \cdot \left( \frac{HB^2}{T^2} + \frac{HB^2}{bT} + \sqrt{\frac{HB^2 L^*}{bT}} \right)$$

*Similarly, there exists $\ell(w; z)$ such that for any algorithm that uses $T$ minibatch stochastic gradients of size $b$, there exists a distribution over $z$ such that $\ell$ and $L$ satisfy Assumptions 1 and 3 (and, in fact, $L$ is $\lambda$-strongly convex), and for universal constants $c, c'$, the algorithm's output will have error at least*

$$\mathbb{E}L(\hat{w}) - L^* \geq c \cdot \left( \Delta \exp\left( -c' \cdot \frac{\sqrt{\lambda}T}{\sqrt{H}} \right) + \Delta \cdot \mathbb{1}_{bT \leq \frac{H}{2\lambda}} + \min\left\{ \frac{HL^*}{\lambda bT}, \Delta \right\} \right)$$

It is easy to see that this precisely matches our algorithm's guarantee, Theorem 1, under Assumptions 1 and 2, and therefore our algorithm is optimal in that setting. Under Assumptions 1 and 3, this lower bound and the upper bound, Theorem 3, nearly match, with the only difference being the terms $\exp\left(-\frac{c' \cdot \lambda bT}{H}\right)$ in the upper bound versus $\mathbb{1}_{bT \leq \frac{H}{2\lambda}}$ in the lower bound. However, this gap is small—for $bT \leq \frac{H}{2\lambda}$, $\exp\left(-\frac{c' \cdot \lambda bT}{H}\right)$, is at most 1, so the upper bound is within a constant factor of the lower bound. For $bT > \frac{H}{2\lambda}$, $\exp\left(-\frac{c' \cdot \lambda bT}{H}\right)$ is obviously more than a constant factor larger than $\mathbb{1}_{bT \leq \frac{H}{2\lambda}} = 0$, but it is nevertheless exponentially small, so the gap between the upper and lower bound is still nearly negligible.

We note that because Theorem 4 applies to any learning rule that uses $n = bT$ i.i.d. samples, not just first-order methods, if the second and third terms of our algorithms' guarantees are larger than the first terms, then our methods are actually optimal amongst all learning rules. Therefore, for small enough $b$, our first-order algorithm is just as good, in the worst case, as any other method including, for example, exact (regularized) empirical risk minimization.

Finally, in the special case that $b = 1$, while it is true that our minibatch accelerated SGD algorithm is optimal—even amongst all learning rules that use $n = bT = T$ samples, it is also the case that plain old SGD is *also* optimal, which guarantees under Assumptions 1 and 2 [10]

$$\mathbb{E}L(\hat{w}) - L^* \leq c \cdot \left( \frac{HB^2}{T} + \sqrt{\frac{HB^2 L^*}{bT}} \right) \tag{11}$$

matching the lower bound, Theorem 4, with $n = bT = T$. In other words, the novelty and advantage of our method appears primarily through in how it leverages minibatches to achieve better performance and a smaller parallel runtime, as we will now discuss.

## 6   The Minibatch Parallelization Speedup

Previously, we showed that our algorithms are optimal in terms of their upper bounds on the error as a function of $b$ and $L^*$. Here, we consider the related question of the algorithm's runtime. Since it is easy to parallelize the computation of minibatch stochastic gradients of size $b$, the total runtime of a minibatch first-order method scales in direct proportion to $T$, but may grow much more slowly with $b$—and it may not grow at all if $b$ parallel computers are available. Specifically, if $M$ computing cores or devices are available to parallelize the minibatch computations, then any minibatch first-order algorithm's runtime would scale with

$$\text{Runtime} \propto T \times \left\lceil \frac{b}{M} \right\rceil \times \text{Time to compute } \nabla \ell(w; z) \tag{12}$$

For this reason, it is natural to ask to what extent we can reduce the runtime without hurting performance, i.e. how much we can reduce the number of iterations, $T$, by increasing the minibatch size, $b$. To answer this question, it will be convenient to rewrite our guarantees in Theorems 1 and 3 by fixing the error, $\epsilon$, and asking how large $T$ must be in order to guarantee error $\epsilon$. Under Assumptions 1 and 2, and Assumptions 1 and 3, respectively, this is (ignoring constants)

$$T(\epsilon) = \sqrt{\frac{HB^2}{\epsilon}} + \frac{1}{b}\left(\frac{HB^2}{\epsilon} + \frac{HB^2 L^*}{\epsilon^2}\right) \tag{13}$$

$$T(\epsilon) = \sqrt{\frac{H}{\lambda}} \log \frac{\Delta}{\epsilon} + \frac{1}{b}\left(\frac{H}{\lambda} \log \frac{\Delta}{\epsilon} + \frac{HL^*}{\lambda \epsilon}\right) \tag{14}$$

Written this way, it is easy to see that the number of iterations needed by our algorithm to reach accuracy $\epsilon$ decreases linearly with the minibatch size $b$ up until the first term becomes larger than the second and third terms. Although it may not be practical to fully parallelize the computation of the minibatches across $b$ workers for large $b$, this nevertheless represents a substantial potential speedup with absolutely no cost to the algorithm's theoretical guarantees [10, 11], which continues until

$$b \geq \sqrt{\frac{HB^2}{\epsilon}} + \frac{\sqrt{HB^2}L^*}{\epsilon^{3/2}} \quad \text{and} \quad b \geq \sqrt{\frac{H}{\lambda}} + \frac{\sqrt{H}L^*}{\epsilon\sqrt{\lambda} \log \frac{\Delta}{\epsilon}} \tag{15}$$

Once $b$ passes these thresholds, no more improvement is possible by increasing the minibatch size, and the iteration complexity is dominated by the first term, which corresponds to the time needed to reach $\epsilon$ error using exact accelerated gradient descent [28]. In other words, once the minibatch size is this large, the algorithm performs essentially the same as if there were no noise in the gradients.

Our algorithms' speedup from minibatching improves significantly over previous results. As mentioned in the previous section, while SGD can attain the same error as our method using the same total number of samples, $n = bT$, it can only do so with $b = 1$, and the number of iterations it requires to attain error $\epsilon$ under Assumptions 1 and 2 is (ignoring constants) [10]

$$T(\epsilon) = \frac{HB^2}{\epsilon} + \frac{1}{b}\frac{HB^2 L^*}{\epsilon^2} \tag{16}$$

Comparing this with our method (13), we see that SGD always requires at least as many iterations, and quadratically more for large $b$. In fact, in the case $L^* = 0$, SGD sees no speedup at all from minibatching, whereas our method can be sped up substantially.

Improving over SGD, the minibatch accelerated SGD algorithm of Cotter et al. [10] requires under Assumptions 1 and 2 (ignoring constants and log factors)

$$T(\epsilon) = \sqrt{\frac{HB^2}{\epsilon}} + \frac{1}{\sqrt{b}}\frac{HB^2}{\epsilon} + \frac{1}{b}\frac{HB^2 L^*}{\epsilon^2} \tag{17}$$

Therefore, their analysis exhibits three regimes rather than our two: first a linear $1/b$ speedup for $b \leq L^{*2}/\epsilon^2$, then a $1/\sqrt{b}$ speedup for

$$\frac{L^{*2}}{\epsilon^2} \leq b \leq \frac{HB^2}{\epsilon} + \frac{\sqrt{HB^2}L^*}{\epsilon^{3/2}} \tag{18}$$

and finally no speedup for larger $b$. Therefore, even under the favorable condition that $L^* = 0$, Cotter et al.'s method has a sublinear parallelization speedup from minibatching, and consequently, their method can result in a substantially smaller speedup than ours for any particular minibatch size. The minibatch size they need to reach error $\epsilon$ in $\sqrt{HB^2/\epsilon}$ iterations—the optimal number for first-order methods, which corresponds to exact accelerated gradient descent—is the righthand side of (18), which can be larger by a factor of as much as $\sqrt{HB^2/\epsilon}$.

Similarly, under Assumptions 1 and 3, and in the special case of least squares problems, the algorithm and analysis of Liu and Belkin [19] exhibits a similar sublinear speedup from minibatching:

$$T(\epsilon) = \sqrt{\frac{H}{\lambda}} \log \frac{\Delta}{\epsilon} + \frac{1}{\sqrt{b}} \frac{H}{\lambda} \log \frac{\Delta}{\epsilon} + \frac{1}{b} \frac{HL^*}{\lambda \epsilon} \tag{19}$$

Like with Cotter et al.'s analysis, compared with our method, which enjoys a linear speedup all the way up to the critical minibatch size, we see that Liu and Belkin's algorithm has only a $1/\sqrt{b}$ speedup in some regimes, so it can require much larger minibatches to match accelerated gradient descent.

## 7 Stochastic Optimization with Bounded Variance at the Optimum

So far, we have considered optimizing objectives where the instantaneous losses are non-negative and the value of the minimum of the expected loss is bounded and small, but in other contexts we may want to understand the complexity of optimization in terms of bounds on the variance of the stochastic gradients. In particular, it is increasingly common in the optimization literature to assume a bound on the variance of the stochastic gradient estimates at the minimizer of the objective, i.e. $\mathbb{E}\|\nabla\ell(w^*; z)\|^2 \leq \sigma_*^2$.

In Appendix E, we extend our analysis to this setting, with $\sigma_*^2$ essentially replacing the quantity $2HL^*$ in all of our results so far. The main result is Theorem 5, in which we prove that in this setting our algorithm guarantees error

$$\mathbb{E}L(w_T^{\mathrm{ag}}) - L^* \leq c \cdot \left( \frac{HB^2}{T^2} + \frac{HB^2}{bT} + \frac{\sigma_* B}{\sqrt{bT}} \right) \tag{20}$$

in the convex setting, and its modification with Algorithm 2 guarantees error

$$\mathbb{E}L(\hat{w}) - L^* \leq c \cdot \left( \Delta \exp\left( -\frac{c'\sqrt{\lambda}T}{\sqrt{H}} \right) + \Delta \exp\left( -\frac{c'\lambda bT}{H} \right) + \frac{\sigma_*^2}{\lambda bT} \right) \tag{21}$$

with the relaxed notion of strong convexity from Assumption 3. We also show lower bounds in Corollary 2 confirming that our method is minimax optimal under these conditions.

## 8 Conclusion

We proposed and analyzed a minibatch accelerated SGD algorithm for optimizing objectives whose minimum value is near zero. We show that our method is simultaneously optimal with respect to the minibatch size, $b$, and the minimum of the loss, $L^*$, which improves over previous results including Cotter et al. [10] and Liu and Belkin [19] which were optimal with respect to $L^*$ but not $b$, and Lan [17] which was optimal with respect to $b$ but not $L^*$. In Section 6, we describe how our method's improvements over prior work, which takes the form of a better dependence on the minibatch size, $b$, translates into the potential for a substantial reduction in the runtime via parallelizing the computation of the minibatch stochastic gradients. Finally, we extend our results to the closely related setting where the $L^*$ bound is replaced by a bound on the variance of the stochastic gradients at the point $w^*$, specifically, and we tightly characterize the minimax optimal rates in this setting. Our algorithm and analysis is of particular interest in the context of training machine learning models in the "interpolation"/"realizable"/"overparametrized" setting, where there exist parameters that exactly or nearly minimize the training and/or population loss, i.e. $L^*$ and $\sigma_*$ are small.

A shortcoming of our method is that its implementation, specifically setting the stepsizes, depends on potentially unknown quantities such as $L^*$ and $B$, and on the time horizon $T$. The dependence on $T$ is not a particularly serious problem because it is straightforward to convert our method to an

anytime algorithm using the classic "doubling trick", although this is not very practical and it would be interesting to develop an anytime variant of Algorithm 1.

On the other hand, the requirement of knowing $L^*$ and $B$ to implement the algorithm is a trickier issue. This problem is not unique to our method, and most of the accelerated stochastic first-order methods that we are aware of, including the work of Lan [17] and Cotter et al. [10], use these parameters to choose stepsizes and momentum parameters. While these quantities are generally unknown, our algorithm only needs an upper bound on them, so for any known upper bound $\tilde{L}^* \geq L^*$ and $\tilde{B} \geq B$, our algorithm can be implemented using the estimates $\tilde{L}^*$ and $\tilde{B}$, with a corresponding degradation in the guarantee depending on how tight the upper bounds are. Furthermore, when applying stochastic first-order algorithms in practice, one typically sets stepsize parameters via cross-validation rather than according to the theoretical prescriptions, so this may not be a big issue in practice.

## Acknowledgments and Disclosure of Funding

We thank Ohad Shamir for several helpful discussions in the process of preparing this article, and also George Lan for a conversation about optimization with bounded $\sigma_*$. BW is supported by a Google Research PhD Fellowship, and this work was also supported by NSF-CCF/BSF award 1718970/2016741, and was done as part of the NSF-Simons Funded Collaboration on the Foundations of Deep Learning (`https://deepfoundations.ai/`).

Additional revenues related to this work: BW was an intern at Google Research. NS was a paid consultant at Google on issues related to the project, and was also hosted by Microsoft.

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
