# A Proof of Theorem 1

**Lemma 1** (c.f. Lemma 1 [17])**.** *Let $w_{t+1}$, $w_t$, and $w_t^{md}$ be updated as in Algorithm 1. Then for any $w \in \{w : \|w\| \leq B\}$*

$$\gamma_t \langle g_t(w_t^{md}), w_{t+1} - w_t^{md} \rangle$$
$$\leq \gamma_t \langle g_t(w_t^{md}), w - w_t^{md} \rangle + \frac{1}{2}\|w - w_t\|^2 - \frac{1}{2}\|w - w_{t+1}\|^2 - \frac{1}{2}\|w_{t+1} - w_t\|^2$$

*Proof.* First, we show that

$$w_{t+1} = \arg\min_{w:\|w\|\leq B} \gamma_t \langle g_t(w_t^{\mathrm{md}}), w - w_t^{\mathrm{md}} \rangle + \frac{1}{2}\|w - w_t\|^2 \tag{22}$$

Let $\hat{w}$ be this arg min, which is unique since the objective is strongly convex. The KKT optimality conditions for $\hat{w}$ are that there exists $\lambda$ such that

$$\|\hat{w}\| \leq B \tag{23}$$
$$\lambda \geq 0 \tag{24}$$
$$\lambda(\|\hat{w}\| - B) = 0 \tag{25}$$
$$\gamma_t g_t(w_t^{\mathrm{md}}) + \hat{w} - w_t + \lambda\hat{w} = 0 \iff \hat{w} = \frac{w_t - \gamma_t g_t(w_t^{\mathrm{md}})}{1 + \lambda} \tag{26}$$

Let

$$\lambda = \frac{1}{\min\left\{1, \frac{B}{\|\tilde{w}_{t+1}\|}\right\}} - 1 \tag{27}$$

We will now show that $w_{t+1}$ and this $\lambda$ satisfy these KKT conditions. Since $w_{t+1} = \min\left\{1, \frac{B}{\|\tilde{w}_{t+1}\|}\right\}\tilde{w}_{t+1}$, we have primal feasibility $\|w_{t+1}\| \leq B$. Also, because $\frac{1}{\min\left\{1, \frac{B}{\|\tilde{w}_{t+1}\|}\right\}} \geq \frac{1}{1}$, we have dual feasibility $\lambda \geq 0$. Next, if $\|w_{t+1}\| < B$, then it must be the case that $\min\left\{1, \frac{B}{\|\tilde{w}_{t+1}\|}\right\} = 1$, which implies $\lambda = 0$, which establishes the complementary slackness condition. Finally, we have stationarity because

$$w_{t+1} = \min\left\{1, \frac{B}{\|\tilde{w}_{t+1}\|}\right\}\tilde{w}_{t+1} = \frac{w_t - \gamma_t g_t(w_t^{\mathrm{md}})}{1 + \lambda} \tag{28}$$

From here, we let $p(w) = \gamma_t \langle g_t(w_t^{\mathrm{md}}), w - w_t^{\mathrm{md}} \rangle$, so $w_{t+1} = \arg\min_{w:\|w\|\leq B} p(w) + \frac{1}{2}\|w - w_t\|^2$. The first-order optimality condition for $w_{t+1}$ is that for all $w \in \{w : \|w\| \leq B\}$,

$$\langle \nabla p(w_{t+1}) + w_{t+1} - w_t, w - w_{t+1} \rangle \geq 0 \tag{29}$$

This, combined with the convexity of $p$ implies

$$p(w) + \frac{1}{2}\|w - w_t\|^2$$
$$= p(w) + \frac{1}{2}\|w_{t+1} - w_t\|^2 + \frac{1}{2}\|w - w_{t+1}\|^2 + \langle w_{t+1} - w_t, w - w_{t+1} \rangle \tag{30}$$
$$\geq p(w_{t+1}) + \frac{1}{2}\|w_{t+1} - w_t\|^2 + \frac{1}{2}\|w - w_{t+1}\|^2 + \langle \nabla p(w_{t+1}) + w_{t+1} - w_t, w - w_{t+1} \rangle \tag{31}$$
$$\geq p(w_{t+1}) + \frac{1}{2}\|w_{t+1} - w_t\|^2 + \frac{1}{2}\|w - w_{t+1}\|^2 \tag{32}$$

Substituting the definition of $p$ and rearranging completes the proof. $\qquad\square$

**Lemma 2.** *Let $\ell(\cdot; z)$ be $H$-smooth, convex, and non-negative for each $z$, let the stochastic gradient variance at $w^*$ be bounded $\mathbb{E}\|\nabla\ell(w^*; z) - \nabla L(w^*)\|^2 \leq \sigma_*^2$, and let $g(w_t^{md}) = \frac{1}{b}\sum_{i=1}^{b}\nabla\ell(w_t^{md}; z_i)$ be a minibatch stochastic gradient of size b. Then*

$$\mathbb{E}\|g(w_t^{md}) - \nabla L(w_t^{md})\|^2 \leq \frac{8H^2B^2}{b\beta_t^2} + \frac{8H}{b}\mathbb{E}[L(w_t^{ag}) - L^*] + \frac{4\sigma_*^2}{b}$$

*Proof.* By the independence of the stochastic gradients $\nabla\ell(w_t^{\mathrm{md}}, z_i)$ and the inequality $\|a + b\|^2 \le 2\|a\|^2 + 2\|b\|^2$, we can upper bound

$$\mathbb{E}\|g(w_t^{\mathrm{md}}) - \nabla L(w_t^{\mathrm{md}})\|^2$$

$$= \mathbb{E}\left\|\frac{1}{b}\sum_{i=1}^{b}\nabla\ell(w_t^{\mathrm{md}}; z_i) - \nabla L(w_t^{\mathrm{md}})\right\|^2 \tag{33}$$

$$= \frac{1}{b^2}\sum_{i=1}^{b}\mathbb{E}\|\nabla\ell(w_t^{\mathrm{md}}; z_i) - \nabla L(w_t^{\mathrm{md}})\|^2 \tag{34}$$

$$\le \frac{1}{b}\mathbb{E}\|\nabla\ell(w_t^{\mathrm{md}}; z_1)\|^2 \tag{35}$$

$$\le \frac{2}{b}\mathbb{E}\|\nabla\ell(w_t^{\mathrm{md}}; z_1) - \nabla\ell(w_t^{\mathrm{ag}}; z_1)\|^2 + \frac{2}{b}\mathbb{E}\|\nabla\ell(w_t^{\mathrm{ag}}; z_1)\|^2 \tag{36}$$

$$\le \frac{2H^2}{b}\mathbb{E}\|w_t^{\mathrm{md}} - w_t^{\mathrm{ag}}\|^2 + \frac{4}{b}\mathbb{E}\|\nabla\ell(w_t^{\mathrm{ag}}; z_1) - \nabla\ell(w^*; z_1)\|^2 + \frac{4}{b}\mathbb{E}\|\nabla\ell(w^*; z_1)\|^2 \tag{37}$$

For the final inequality, we used that $\ell(\cdot; z)$ is $H$-smooth, so $\nabla\ell(\cdot; z)$ is $H$-Lipschitz.

For the first term on the right hand side, we note that due to the algorithm's projections, all of the iterates $w_t^{\mathrm{md}}$, $w_t^{\mathrm{ag}}$, and $w_t$ lie within the set $\{w : \|w\| \le B\}$. Therefore,

$$w_t^{\mathrm{md}} = \beta_t^{-1}w_t + (1 - \beta_t^{-1})w_t^{\mathrm{ag}} \implies \|w_t^{\mathrm{md}} - w_t^{\mathrm{ag}}\| = \beta_t^{-1}\|w_t - w_t^{\mathrm{ag}}\| \le 2B\beta_t^{-1} \tag{38}$$

For the second term, we apply [Theorem 2.1.5 27]:

$$\mathbb{E}\|\nabla\ell(w_t^{\mathrm{ag}}; z_1) - \nabla\ell(w^*; z_1)\|^2$$
$$\le 2H\mathbb{E}\big[\ell(w_t^{\mathrm{ag}}; z_1) - \ell(w^*; z_1) - \langle\nabla\ell(w^*; z_1), w_t^{\mathrm{ag}} - w^*\rangle\big] \tag{39}$$
$$= 2H\mathbb{E}\big[L(w_t^{\mathrm{ag}}) - L^*\big] \tag{40}$$

For the third term, we use the variance bound at $w^*$:

$$\mathbb{E}\|\nabla\ell(w^*; z_1)\|^2 = \mathbb{E}\|\nabla\ell(w^*; z_1) - \nabla L(w^*)\|^2 \le \sigma_*^2 \tag{41}$$

Combining these with (37) completes the proof. $\qquad\square$

**Lemma 3.** *Let $\ell(\cdot; z)$ be $H$-smooth and non-negative for all $z$ and let $L^* = \min_w L(w)$. Then*

$$\mathbb{E}\|\nabla\ell(w^*; z)\|^2 = \mathbb{E}\|\nabla\ell(w^*; z) - \nabla L(w^*)\|^2 \le 2HL^*$$

*Proof.* This follows almost immediately from [Theorem 2.1.5 27]. For each $z$, let $w_z^* \in \arg\min_w \ell(w; z)$, then

$$\mathbb{E}\|\nabla\ell(w^*; z)\|^2 = \mathbb{E}\|\nabla\ell(w^*; z) - \nabla\ell(w_z^*; z)\|^2 \tag{42}$$
$$\le 2H\mathbb{E}[\ell(w^*; z) - \ell(w_z^*; z) - \langle\nabla\ell(w_z^*; z), w^* - w_z^*\rangle] \tag{43}$$
$$= 2HL^* - 2H\mathbb{E}\ell(w_z^*; z) \tag{44}$$
$$\le 2HL^* \tag{45}$$

For the final inequality, we used that $\ell$ is non-negative. $\qquad\square$

**Theorem 1.** *Let $\ell$ and $L$ satisfy Assumptions 1 and 2, then Algorithm 1 guarantees for a universal constant $c$*

$$\mathbb{E}L(w_T^{ag}) - L^* \le c \cdot \left(\frac{HB^2}{T^2} + \frac{HB^2}{bT} + \sqrt{\frac{HB^2L^*}{bT}}\right)$$

*Proof.* This proof is based on similar ideas as the proof of Lemma 5 and Theorem 2 due to Lan [17]. The key difference is that Lan considers a setting in which the variance of the stochastic gradients are uniformly bounded, while in our setting, we do not directly assume any bound on this quantity.

Let $d_t = w_{t+1} - w_t$, it can be easily seen that

$$w_{t+1}^{\mathrm{ag}} - w_t^{\mathrm{md}} = \beta_t^{-1}w_{t+1} + (1 - \beta_t^{-1})w_t^{\mathrm{ag}} - w_t^{\mathrm{md}} = \beta_t^{-1}d_t \tag{46}$$

The above observation, along with the $H$-smoothness of $L$ implies

$$\beta_t \gamma_t L(w_{t+1}^{\mathrm{ag}}) \leq \beta_t \gamma_t \left[ L(w_t^{\mathrm{md}}) + \left\langle \nabla L(w_t^{\mathrm{md}}), \, w_{t+1}^{\mathrm{ag}} - w_t^{\mathrm{md}} \right\rangle + \frac{H}{2} \left\| w_{t+1}^{\mathrm{ag}} - w_t^{\mathrm{md}} \right\|^2 \right] \tag{47}$$

$$= \beta_t \gamma_t \left[ L(w_t^{\mathrm{md}}) + \left\langle \nabla L(w_t^{\mathrm{md}}), \, w_{t+1}^{\mathrm{ag}} - w_t^{\mathrm{md}} \right\rangle \right] + \frac{H\gamma_t}{2\beta_t} \|d_t\|^2 \tag{48}$$

Using the convexity of $L$, we can upper bound:

$$\beta_t \gamma_t \left[ L(w_t^{\mathrm{md}}) + \left\langle \nabla L(w_t^{\mathrm{md}}), \, w_{t+1}^{\mathrm{ag}} - w_t^{\mathrm{md}} \right\rangle \right]$$
$$= \beta_t \gamma_t \left[ L(w_t^{\mathrm{md}}) + \left\langle \nabla L(w_t^{\mathrm{md}}), \, \beta_t^{-1} w_{t+1} + (1 - \beta_t^{-1}) w_t^{\mathrm{ag}} - w_t^{\mathrm{md}} \right\rangle \right] \tag{49}$$
$$= (\beta_t - 1)\gamma_t \left[ L(w_t^{\mathrm{md}}) + \left\langle \nabla L(w_t^{\mathrm{md}}), \, w_t^{\mathrm{ag}} - w_t^{\mathrm{md}} \right\rangle \right]$$
$$\quad + \gamma_t \left[ L(w_t^{\mathrm{md}}) + \left\langle \nabla L(w_t^{\mathrm{md}}), \, w_{t+1} - w_t^{\mathrm{md}} \right\rangle \right] \tag{50}$$
$$\leq (\beta_t - 1)\gamma_t L(w_t^{\mathrm{ag}}) + \gamma_t \left[ L(w_t^{\mathrm{md}}) + \left\langle g_t(w_t^{\mathrm{md}}), \, w_{t+1} - w_t^{\mathrm{md}} \right\rangle \right] - \gamma_t \left\langle \delta_t, \, w_{t+1} - w_t^{\mathrm{md}} \right\rangle \tag{51}$$

where $\delta_t := g_t(w_t^{\mathrm{md}}) - \nabla L(w_t^{\mathrm{md}})$. We now apply Lemma 1 to conclude that for any $w \in \{w : \|w\| \leq B\}$

$$\gamma_t \left\langle g_t(w_t^{\mathrm{md}}), \, w_{t+1} - w_t^{\mathrm{md}} \right\rangle$$
$$\leq \gamma_t \left\langle g_t(w_t^{\mathrm{md}}), \, w - w_t^{\mathrm{md}} \right\rangle + \frac{1}{2}\|w - w_t\|^2 - \frac{1}{2}\|w - w_{t+1}\|^2 - \frac{1}{2}\|w_{t+1} - w_t\|^2 \tag{52}$$

Because there exists a minimizer of $L$ with norm at most $B$, we can apply this with $w = w^* \in \arg\min_{w:\|w\| \leq B} L(w)$. This, plus the convexity of $L$ allows us to upper bound the second term in (51) as

$$\gamma_t L(w_t^{\mathrm{md}}) + \gamma_t \left\langle g_t(w_t^{\mathrm{md}}), \, w_{t+1} - w_t^{\mathrm{md}} \right\rangle$$
$$= \gamma_t L(w_t^{\mathrm{md}}) + \gamma_t \left\langle g_t(w_t^{\mathrm{md}}), \, w^* - w_t^{\mathrm{md}} \right\rangle$$
$$\quad + \frac{1}{2}\|w^* - w_t\|^2 - \frac{1}{2}\|w^* - w_{t+1}\|^2 - \frac{1}{2}\|w_{t+1} - w_t\|^2 \tag{53}$$
$$= \gamma_t L(w_t^{\mathrm{md}}) + \gamma_t \left\langle \nabla L(w_t^{\mathrm{md}}), \, w^* - w_t^{\mathrm{md}} \right\rangle + \gamma_t \left\langle \delta_t, \, w^* - w_t^{\mathrm{md}} \right\rangle$$
$$\quad + \frac{1}{2}\|w^* - w_t\|^2 - \frac{1}{2}\|w^* - w_{t+1}\|^2 - \frac{1}{2}\|w_{t+1} - w_t\|^2 \tag{54}$$
$$\leq \gamma_t L^* + \gamma_t \left\langle \delta_t, \, w^* - w_t^{\mathrm{md}} \right\rangle + \frac{1}{2}\|w^* - w_t\|^2 - \frac{1}{2}\|w^* - w_{t+1}\|^2 - \frac{1}{2}\|w_{t+1} - w_t\|^2 \tag{55}$$

Therefore, returning to (51), we conclude that

$$\beta_t \gamma_t \left[ L(w_t^{\mathrm{md}}) + \left\langle \nabla L(w_t^{\mathrm{md}}), \, w_{t+1}^{\mathrm{ag}} - w_t^{\mathrm{md}} \right\rangle \right] \leq (\beta_t - 1)\gamma_t L(w_t^{\mathrm{ag}}) + \gamma_t L^*$$
$$+ \gamma_t \left\langle \delta_t, \, w^* - w_{t+1} \right\rangle + \frac{1}{2}\left( -\|w_{t+1} - w_t\|^2 + \|w_t - w^*\|^2 - \|w_{t+1} - w^*\|^2 \right) \tag{56}$$

Plugging this back into (48) and subtracting $\beta_t \gamma_t L^*$ from both sides, this implies

$$\beta_t \gamma_t \left[ L(w_{t+1}^{\mathrm{ag}}) - L^* \right] \leq (\beta_t - 1)\gamma_t \left[ L(w_t^{\mathrm{ag}}) - L^* \right] + \frac{1}{2}\|w_t - w^*\|^2 - \frac{1}{2}\|w_{t+1} - w^*\|^2$$
$$+ \frac{H\gamma_t - \beta_t}{2\beta_t}\|w_t - w_{t+1}\|^2 + \gamma_t \left\langle \delta_t, \, w^* - w_{t+1} \right\rangle \tag{57}$$

$$= (\beta_t - 1)\gamma_t \left[ L(w_t^{\mathrm{ag}}) - L^* \right] + \frac{1}{2}\|w_t - w^*\|^2 - \frac{1}{2}\|w_{t+1} - w^*\|^2$$
$$+ \frac{H\gamma_t - \beta_t}{2\beta_t}\|w_t - w_{t+1}\|^2 + \gamma_t \left\langle \delta_t, \, w_t - w_{t+1} \right\rangle + \gamma_t \left\langle \delta_t, \, w^* - w_t \right\rangle \tag{58}$$

$$\leq (\beta_t - 1)\gamma_t \left[ L(w_t^{\mathrm{ag}}) - L^* \right] + \frac{1}{2}\|w_t - w^*\|^2 - \frac{1}{2}\|w_{t+1} - w^*\|^2$$
$$+ \frac{H\gamma_t - \beta_t}{2\beta_t}\|w_t - w_{t+1}\|^2 + \gamma_t \|\delta_t\| \|w_t - w_{t+1}\| + \gamma_t \left\langle \delta_t, \, w^* - w_t \right\rangle \tag{59}$$

Because $\beta_t = 1 + \frac{t}{6} > \frac{1+t}{6} \geq 2H\gamma_t$, the first two terms on the second line of the right hand side are a quadratic polynomial of the form $-\frac{a}{2}y^2 + by$ (here, $y$ corresponds to $\|w_t - w_{t+1}\|$), which can be upper bounded by $-\frac{a}{2}y^2 + by \leq \max_y\left\{-\frac{a}{2}y^2 + by\right\} = \frac{b^2}{2a}$. We conclude

$$\beta_t\gamma_t\left[L(w_{t+1}^{\text{ag}}) - L^*\right] \leq (\beta_t - 1)\gamma_t\left[L(w_t^{\text{ag}}) - L^*\right] + \frac{1}{2}\|w_t - w^*\|^2 - \frac{1}{2}\|w_{t+1} - w^*\|^2$$
$$+ \frac{\beta_t\gamma_t^2}{2(\beta_t - H\gamma_t)}\|\delta_t\|^2 + \gamma_t\langle\delta_t, w^* - w_t\rangle \tag{60}$$

$$\leq (\beta_t - 1)\gamma_t\left[L(w_t^{\text{ag}}) - L^*\right] + \frac{1}{2}\|w_t - w^*\|^2 - \frac{1}{2}\|w_{t+1} - w^*\|^2$$
$$+ \gamma_t^2\|\delta_t\|^2 + \gamma_t\langle\delta_t, w^* - w_t\rangle \tag{61}$$

Taking the expectation of both sides, and noting that the noise in the $t^{\text{th}}$ stochastic gradient estimate, $g_t(w_t^{\text{md}})$, is independent of $w_t$ so that $\mathbb{E}\langle\delta_t, w^* - w_t\rangle = 0$, we have

$$\beta_t\gamma_t\mathbb{E}\left[L(w_{t+1}^{\text{ag}}) - L^*\right] \leq (\beta_t - 1)\gamma_t\mathbb{E}\left[L(w_t^{\text{ag}}) - L^*\right] + \frac{1}{2}\mathbb{E}\|w_t - w^*\|^2 - \frac{1}{2}\mathbb{E}\|w_{t+1} - w^*\|^2$$
$$+ \gamma_t^2\mathbb{E}\|g_t(w_t^{\text{md}}) - \nabla F(w_t^{\text{md}})\|^2 \tag{62}$$

We now use Lemma 2 to bound the variance of the minibatch stochastic gradient at $w_t^{\text{md}}$, which yields

$$\beta_t\gamma_t\mathbb{E}\left[L(w_{t+1}^{\text{ag}}) - L^*\right]$$
$$\leq (\beta_t - 1)\gamma_t\mathbb{E}\left[L(w_t^{\text{ag}}) - L^*\right] + \frac{1}{2}\mathbb{E}\|w_t - w^*\|^2 - \frac{1}{2}\mathbb{E}\|w_{t+1} - w^*\|^2$$
$$+ \frac{8H^2B^2\gamma_t^2}{b\beta_t^2} + \frac{8H\gamma_t^2}{b}\mathbb{E}\left[L(w_t^{\text{ag}}) - L^*\right] + \frac{4\sigma_*^2\gamma_t^2}{b} \tag{63}$$

$$\leq \left(\beta_t - 1 + \frac{8H\gamma_t}{b}\right)\gamma_t\mathbb{E}\left[L(w_t^{\text{ag}}) - L^*\right] + \frac{1}{2}\mathbb{E}\|w_t - w^*\|^2 - \frac{1}{2}\mathbb{E}\|w_{t+1} - w^*\|^2$$
$$+ \frac{8H^2B^2\gamma_t^2}{b\beta_t^2} + \frac{4\sigma_*^2\gamma_t^2}{b} \tag{64}$$

From here, we recall that

$$\beta_t = 1 + \frac{t}{6} \tag{65}$$
$$\gamma_t = \gamma(t + 1) \tag{66}$$
$$\gamma \leq \min\left\{\frac{1}{12H}, \frac{b}{24H(T+1)}\right\} \tag{67}$$

This ensures that $\beta_t \geq 1$ and $2H\gamma_t \leq \beta_t$ for all $t$. Furthermore, for $0 \leq t \leq T - 1$

$$\left(\beta_{t+1} - 1 + \frac{8H\gamma_{t+1}}{b}\right)\gamma_{t+1} - \beta_t\gamma_t \tag{68}$$

$$= \left(\beta_t - \frac{5}{6} + \frac{8H\gamma_{t+1}}{b}\right)\gamma(t + 2) - \beta_t\gamma(t + 1) \tag{69}$$

$$= \gamma\left(1 + \frac{t}{6} - \frac{5(t+2)}{6} + \frac{8H\gamma(t+2)^2}{b}\right) \tag{70}$$

$$= \gamma\left(-\frac{2}{3} - \frac{2t}{3} + \frac{(t+2)}{3} \cdot \frac{24H(t+2)\gamma}{b}\right) \tag{71}$$

$$\leq \gamma\left(-\frac{t}{3}\right) \leq 0 \tag{72}$$

Therefore, $\left(\beta_{t+1} - 1 + \frac{8H\gamma_{t+1}}{b}\right)\gamma_{t+1} \leq \beta_t\gamma_t$ for all $0 \leq t \leq T-1$. We can now unroll the recurrence (64) to conclude

$$\left(\beta_T - 1 + \frac{8H\gamma_T}{b}\right)\gamma_T\mathbb{E}\left[L(w_T^{\mathrm{ag}}) - L^*\right]$$

$$\leq \beta_{T-1}\gamma_{T-1}\mathbb{E}\left[L(w_T^{\mathrm{ag}}) - L^*\right] \tag{73}$$

$$\leq \left(\beta_{T-1} - 1 + \frac{8H\gamma_{T-1}}{b}\right)\gamma_{T-1}\mathbb{E}\left[L(w_{T-1}^{\mathrm{ag}}) - L^*\right] + \frac{1}{2}\mathbb{E}\|w_{T-1} - w^*\|^2 - \frac{1}{2}\mathbb{E}\|w_T - w^*\|^2$$

$$+ \frac{8H^2B^2\gamma_{T-1}^2}{b\beta_{T-1}^2} + \frac{4\sigma_*^2\gamma_{T-1}^2}{b} \tag{74}$$

$$\vdots \tag{75}$$

$$\leq \frac{1}{2}\mathbb{E}\|w_0 - w^*\|^2 + \sum_{t=0}^{T-1}\left[\frac{8H^2B^2\gamma_t^2}{b\beta_t^2} + \frac{4\sigma_*^2\gamma_t^2}{b}\right] \tag{76}$$

$$\leq \frac{B^2}{2} + \sum_{t=0}^{T-1}\left[\frac{288H^2B^2\gamma^2(t+1)^2}{b(t+6)^2} + \frac{4\sigma_*^2\gamma^2(t+1)^2}{b}\right] \tag{77}$$

$$\leq \frac{B^2}{2} + \frac{288H^2B^2\gamma^2T}{b} + \frac{4\sigma_*^2\gamma^2T^3}{b} \tag{78}$$

In addition, we have

$$\left(\beta_T - 1 + \frac{8H\gamma_T}{b}\right)\gamma_T = \left(\frac{T}{6} + \frac{8H\gamma(T+1)}{b}\right)\gamma(T+1) \geq \frac{\gamma T^2}{6} \tag{79}$$

Therefore,

$$\mathbb{E}\left[L(w_T^{\mathrm{ag}}) - L^*\right] \leq \frac{3B^2}{\gamma T^2} + \frac{1728H^2B^2}{bT}\gamma + \frac{24\sigma_*^2 T}{b}\gamma \tag{80}$$

With our choice of[2]

$$\gamma = \min\left\{\frac{1}{12H}, \frac{b}{24H(T+1)}, \sqrt{\frac{\frac{B^2}{T^2}}{\frac{\sigma_*^2 T}{b}}}\right\} \tag{81}$$

this means

$$\mathbb{E}\left[L(w_T^{\mathrm{ag}}) - L^*\right]$$

$$\leq \frac{3B^2}{T^2\min\left\{\frac{1}{12H}, \frac{b}{24H(T+1)}, \sqrt{\frac{\frac{B^2}{T^2}}{\frac{\sigma_*^2 T}{b}}}\right\}} + \frac{72HB^2}{T(T+1)} + \frac{24\sigma_*B}{\sqrt{bT}} \tag{82}$$

$$\leq \frac{36HB^2}{T^2} + \frac{72HB^2(T+1)}{bT^2} + \frac{3\sigma_*B}{\sqrt{bT}} + \frac{72HB^2}{T(T+1)} + \frac{24\sigma_*B}{\sqrt{bT}} \tag{83}$$

$$\leq \frac{108HB^2}{T^2} + \frac{144HB^2}{bT} + \frac{27\sigma_*B}{\sqrt{bT}} \tag{84}$$

We complete the proof by applying Lemma 3, which shows that $\mathbb{E}\|\nabla\ell(w^*; z) - \nabla L(w^*)\|^2 \leq \sigma_*^2$ for $\sigma_*^2 = 2HL^*$. $\qquad\square$

# B  Proof and Additional Applications of Theorem 2

**Theorem 2.** *For any algorithm $\mathcal{A}$ and $\theta > 1$, $\mathsf{GC2Cvx}(\mathcal{A}, \theta)$ defined in Algorithm 2 guarantees*

$$\mathsf{Time}_\lambda(\epsilon, \Delta, \psi, \mathcal{L}, \mathsf{GC2Cvx}(\mathcal{A}, \theta)) \leq \sum_{t=1}^{\lceil\log_\theta\frac{\Delta}{\epsilon}\rceil}\mathsf{Time}\left(\theta^{-t}\Delta, \theta^{1-t}\frac{\Delta}{\lambda}, \psi, \mathcal{L}, \mathcal{A}\right)$$

---

[2]Algorithm 1 defines $\gamma$ in terms of $HL^*$ rather than $\sigma_*^2$. Later in this proof, we apply Lemma 3 to bound the variance at $w^*$ by $\sigma_*^2 = 2HL^*$, which justifies this difference.

*Proof.* By the definition of $\mathsf{Time}\big(\theta^{-t}\Delta, \theta^{1-t}\frac{\Delta}{\lambda}, \psi, \mathcal{L}, \mathcal{A}\big)$, if $\mathbb{E}\psi(w_{t-1}) \leq \theta^{1-t}\frac{\Delta}{\lambda}$ at each iteration, then $\mathbb{E}L(w_t) - L^* \leq \theta^{-t}\Delta$ for each $t$, and $\mathbb{E}L(w_T) - L^* \leq \theta^{-T}\Delta \leq \epsilon$. We now prove by induction that the condition $\mathbb{E}\psi(w_{t-1}) \leq \theta^{1-t}\frac{\Delta}{\lambda}$ always holds.

As the base case, the $(\lambda, \psi)$-GC implies that

$$\lambda\psi(w_0) \leq L(w_0) - L^* \implies \mathbb{E}\psi(w_0) \leq \frac{\Delta}{\lambda} \tag{85}$$

Now, suppose that for all $t' < t$, $\mathbb{E}\psi(w_{t'}) \leq \theta^{-t'}\frac{\Delta}{\lambda}$. Then, by the definition of $\mathsf{Time}\big(\theta^{-t}\Delta, \theta^{1-t}\frac{\Delta}{\lambda}, \psi, \mathcal{L}, \mathcal{A}\big)$, we have $\mathbb{E}L(w_t) - L^* \leq \theta^{-t}\Delta$ so by the $(\lambda, \psi)$-GC

$$\lambda\psi(w_t) \leq L(w_t) - L^* \implies \mathbb{E}\psi(w_t) \leq \frac{\mathbb{E}L(w_t) - L^*}{\lambda} \leq \theta^{-t}\frac{\Delta}{\lambda} \tag{86}$$

This completes the proof. $\qquad\square$

To better understand Theorem 2, it is useful to consider an few examples:

**Example: Gradient Descent for Lipschitz Objectives** Let $\mathcal{L}_G$ be the set of all $G$-Lipschitz, convex objectives, and let $\psi(w) = \frac{1}{2}\|w - w^*\|_2^2$. It is well known that the gradient descent algorithm, which we denote $\mathcal{A}_{GD}$, requires

$$\mathsf{Time}(\epsilon, B^2, \psi, \mathcal{L}_G, \mathcal{A}_{GD}) \leq c \cdot \frac{G^2 B^2}{\epsilon^2} \tag{87}$$

gradients to find an $\epsilon$-suboptimal point, where $c$ is a universal constant. Theorem 2 implies that

$$\mathsf{Time}_\lambda(\epsilon, \Delta, \psi, \mathcal{L}_G, \mathsf{GC2Cvx}(\mathcal{A}_{GD}, e))$$

$$\leq \sum_{t=1}^{\lceil \log\frac{\Delta}{\epsilon} \rceil} \mathsf{Time}\left(e^{-t}\Delta, e^{1-t}\frac{\Delta}{\lambda}, \psi, \mathcal{L}_G, \mathcal{A}_{GD}\right) \tag{88}$$

$$\leq cG^2 \sum_{t=1}^{\lceil \log\frac{\Delta}{\epsilon} \rceil} \frac{e^{1-t}\frac{\Delta}{\lambda}}{e^{-2t}\Delta^2} \tag{89}$$

$$\leq c\frac{e^2 G^2}{(e-1)\lambda\Delta} e^{\lceil \log\frac{\Delta}{\epsilon} \rceil} \tag{90}$$

$$\leq c'\frac{G^2}{\lambda\epsilon} \tag{91}$$

Therefore, our reduction recovers (up to constant factors) the existing guarantee for Lipschitz and strongly convex optimization [25]. We emphasize that this guarantee (91) has nothing to do with gradient descent specifically—for any algorithm $\mathcal{A}$ with

$$\mathsf{Time}(\epsilon, B^2, \psi, \mathcal{L}_G, \mathcal{A}) \leq c \cdot \frac{G^2 B^2}{\epsilon^2}, \tag{92}$$

the modified algorithm $\mathsf{GC2Cvx}(\mathcal{A}, e)$ will have the same rate (91).

**Example: Accelerated SGD for Smooth Objectives** For $\mathcal{L}_H$, the class of convex and $H$-smooth objectives, Lan [17] proposed an algorithm, AC-SA which, for $\psi(w) = \frac{1}{2}\|w - w^*\|^2$, requires

$$\mathsf{Time}(\epsilon, B^2, d_2, \mathcal{L}_H, \mathcal{A}_{AC-SA}) = c \cdot \left(\sqrt{\frac{HB^2}{\epsilon}} + \frac{\sigma^2 B^2}{\epsilon^2}\right) \tag{93}$$

stochastic gradients with variance bounded by $\sigma^2$ to find an $\epsilon$-suboptimal point, which is optimal. In follow-up work Ghadimi and Lan [13] describe a "multi-stage" variant of AC-SA which is optimal for strongly convex objectives. This algorithm closely resembles $\mathsf{GC2Cvx}(\mathcal{A}_{AC-SA}, e)$ with some small differences, and their analysis is what inspired Theorem 2 in the first place. But, in contrast to

their long and fairly complicated analysis, Theorem 2 can be used to prove their guarantee using the following simple computation:

$$\text{Time}_\lambda(\epsilon, \Delta, \psi, \mathcal{L}_H, \text{GC2Cvx}(\mathcal{A}_{AC-SA}, e))$$

$$\leq \sum_{t=1}^{\lceil \log \frac{\Delta}{\epsilon} \rceil} \text{Time}\left( e^{-t}\Delta, e^{1-t}\frac{\Delta}{\lambda}, \psi, \mathcal{L}_H, \mathcal{A}_{AC-SA} \right) \tag{94}$$

$$= c \cdot \left( \sqrt{H} \sum_{t=1}^{\lceil \log \frac{\Delta}{\epsilon} \rceil} \sqrt{\frac{e^{1-t}\frac{\Delta}{\lambda}}{e^{-t}\Delta}} + \sigma^2 \sum_{t=1}^{\lceil \log \frac{\Delta}{\epsilon} \rceil} \frac{e^{1-t}\frac{\Delta}{\lambda}}{e^{-2t}\Delta^2} \right) \tag{95}$$

$$\leq ec \cdot \left( \sqrt{\frac{H}{\lambda}} \left\lceil \log \frac{\Delta}{\epsilon} \right\rceil + \frac{\sigma^2}{\lambda\Delta} \sum_{t=1}^{\lceil \log \frac{\Delta}{\epsilon} \rceil} e^t \right) \tag{96}$$

$$\leq ec \cdot \left( \sqrt{\frac{H}{\lambda}} \left\lceil \log \frac{\Delta}{\epsilon} \right\rceil + \frac{e\sigma^2}{(e-1)\lambda\Delta} \exp\left( \left\lceil \log \frac{\Delta}{\epsilon} \right\rceil \right) \right) \tag{97}$$

$$\leq c' \cdot \left( \sqrt{\frac{H}{\lambda}} \left\lceil \log \frac{\Delta}{\epsilon} \right\rceil + \frac{\sigma^2}{\lambda\epsilon} \right) \tag{98}$$

This is, up to constant factors, the optimal rate for strongly convex objectives, and matches Ghadimi and Lan's analysis.

## C   Proof of Theorem 3

**Theorem 3.** *Let $\ell$ and $L$ satisfy Assumptions 1 and 3, then the output of* GC2Cvx(*Alg 1, e*) *guarantees for universal constants $c, c'$*

$$\mathbb{E}L(\hat{w}) - L^* \leq c \cdot \left( \Delta \exp\left( -\frac{c'\sqrt{\lambda}T}{\sqrt{H}} \right) + \Delta \exp\left( -\frac{c'\lambda bT}{H} \right) + \frac{HL^*}{\lambda bT} \right)$$

*Proof.* Let $\psi(w) = \frac{1}{2} \min_{w^* \in \arg\min_w L(w)} \|w - w^*\|^2$, so $L$ satisfies the $(\lambda, \psi)$-GC. By Theorem 1, Algorithm 1 guarantees that[3]

$$\text{Time}\left( \epsilon, \frac{B^2}{2}, \psi, \mathcal{L}, \text{Alg 1} \right) \leq c \cdot \left( \sqrt{\frac{HB^2}{\epsilon}} + \frac{HB^2}{b\epsilon} + \frac{HB^2L^*}{b\epsilon^2} \right) \tag{99}$$

where, in this case, the "time" refers to the number of iterations, $T$. Applying Theorem 2, this implies

$$\text{Time}_\lambda(\epsilon, \Delta, \psi, \mathcal{L}, \text{GC2Cvx}(\text{Alg 1}, e))$$

$$\leq \sum_{t=1}^{\lceil \log \frac{\Delta}{\epsilon} \rceil} \text{Time}\left( e^{-t}\Delta, e^{1-t}\frac{\Delta}{\lambda}, \psi, \mathcal{L}, \text{Alg 1} \right) \tag{100}$$

$$\leq c \cdot \sum_{t=1}^{\lceil \log \frac{\Delta}{\epsilon} \rceil} \left( \sqrt{\frac{He^{1-t}\frac{\Delta}{\lambda}}{e^{-t}\Delta}} + \frac{He^{1-t}\frac{\Delta}{\lambda}}{be^{-t}\Delta} + \frac{HL^*e^{1-t}\frac{\Delta}{\lambda}}{be^{-2t}\Delta^2} \right) \tag{101}$$

$$= c \cdot \left( \left( \sqrt{\frac{eH}{\lambda}} + \frac{eH}{b\lambda} \right) \left\lceil \log \frac{\Delta}{\epsilon} \right\rceil + \frac{eHL^*}{b\lambda\Delta} \sum_{t=1}^{\lceil \log \frac{\Delta}{\epsilon} \rceil} e^t \right) \tag{102}$$

$$\leq e^3 c \cdot \left( \left( \sqrt{\frac{H}{\lambda}} + \frac{H}{b\lambda} \right) \left\lceil \log \frac{\Delta}{\epsilon} \right\rceil + \frac{HL^*}{b\lambda\epsilon} \right) \tag{103}$$

Solving for $\epsilon$ completes the proof. □

---

[3]Theorem 1, as stated, requires a bound on $\|w^*\|$. However, given $w_0$ with $\psi(w_0) \leq \frac{1}{2}B^2$, there is a minimizer with norm at most $B$ in the shifted coordinate system $w \mapsto w - w_0$.

# D Proof of Theorem 4

**Lemma 4.** *Let $\mu$ be an unknown parameter in $\{\pm a\}$. The output $\hat{\mu}$ of any algorithm which receives as input $k$ i.i.d. samples $x_1, \ldots, x_k \sim \mathcal{N}(\mu, s^2)$ will have mean squared error at least*

$$\max_{\mu \in \{\pm a\}} \mathbb{E}(\hat{\mu} - \mu)^2 \geq \left(1 - \frac{a\sqrt{k}}{s}\right)a^2$$

*Proof.* This lemma is nearly identical to many lower bounds for Gaussian mean estimation. We include the proof to be self-contained and to account for the fact that $\mu$ has only two possible values.

The KL divergence between $k$ i.i.d. samples from $\mathcal{N}(-a, s^2)$ and $\mathcal{N}(a, s^2)$ is

$$D_{KL}(\mathcal{N}(-a, s^2)^{\otimes k} \| \mathcal{N}(a, s^2)^{\otimes k}) = \frac{2ka^2}{s^2} \tag{104}$$

By Pinsker's inequality, the total variation distance between the output of the algorithm if $\mu = -a$ and the output of the algorithm if $\mu = a$ is upper bounded by

$$\delta(\hat{\mu}_{-a}, \hat{\mu}_a) \leq \frac{a\sqrt{k}}{s} \tag{105}$$

Finally, we note that

$$(\hat{\mu} - a)^2 \leq a^2 \implies (\hat{\mu} - (-a))^2 > a^2 \tag{106}$$

and vice versa. Therefore, we conclude that

$$\max_{\mu \in \{\pm a\}} \mathbb{E}(\hat{\mu} - \mu)^2 \geq \left(1 - \frac{a\sqrt{k}}{s}\right)a^2 \tag{107}$$

This completes the proof. $\qquad\square$

**Theorem 4.** *For $\ell(w; (x, y)) = \frac{1}{2}(\langle w, x \rangle - y)^2$ the square loss, for any learning algorithm that takes $n$ samples as input, there exists a distribution over $(x, y)$ pairs such that $\ell$ and $L$ satisfy Assumptions 1 and 2, and for a universal constant $c$, the algorithm's output will have error at least*

$$\mathbb{E}L(\hat{w}) - L^* \geq c \cdot \left(\frac{HB^2}{n} + \sqrt{\frac{HB^2 L^*}{n}}\right)$$

*Similarly, there exists a distribution over $(x, y)$ pairs such that $\ell$ and $L$ satisfy Assumptions 1 and 3 (and, in fact, $L$ is $\lambda$-strongly convex), and for a universal constant $c$, the algorithm's output will have error at least*

$$\mathbb{E}L(\hat{w}) - L^* \geq c \cdot \left(\Delta \cdot \mathbb{1}_{n \leq \frac{H}{2\lambda}} + \min\left\{\frac{HL^*}{\lambda n}, \Delta\right\}\right)$$

*Proof.* We will prove the first terms and the second terms of the lower bounds separately.

**The first terms of each bound** These lower bounds are based on a simple least squares problem in dimension $2n$. The loss is, again,

$$\ell(w; (x; y)) = \frac{1}{2}(\langle w, x \rangle - y)^2 \tag{108}$$

The data distribution is specified in terms of a sign vector $\sigma \in \{\pm 1\}^{2n}$. The $x$ distribution is the uniform distribution over $\{\sqrt{H}e_1, \ldots, \sqrt{H}e_{2n}\}$, and $y|x = \left\langle x, \frac{B}{\sqrt{2n}}\sigma \right\rangle$. Because $\|x\|^2 = H$, it is easy to confirm that $\ell$ is $H$-smooth, convex, and non-negative, so it satisfies Assumption 1. In addition, the expected loss is

$$L(w) = \mathbb{E}_{x,y} \frac{1}{2}(\langle w, x \rangle - y)^2 = \frac{1}{4n} \sum_{i=1}^{2n} \left(\sqrt{H}w_i - \frac{\sqrt{H}B}{\sqrt{2n}}\sigma_i\right)^2 \tag{109}$$

It is easy to see that $L$ is minimized at the point $w^* = \frac{B}{\sqrt{2n}}\sigma$, which has norm $B$ and that $L(w^*) = L^* = 0$. Therefore, $L$ satisfies Assumption 2.

Alternatively, $L(0) - L^* = \frac{HB^2}{4n}$, so choosing $B^2 = \frac{4n\Delta}{H}$ ensures that $L(0) - L^* \leq \Delta$. Also, $L$ is $\frac{H}{2n}$-strongly convex, so it satisfies Assumption 3 as long as $n \leq \frac{H}{2\lambda}$.

Finally, any algorithm which sees $n$ samples from the distribution will have received no information whatsoever about the $\geq n$ coordinates of the sign vector $\sigma$ that were not involved in the sample. Therefore, for any algorithm, there is a setting of $\sigma$ such that $\mathbb{P}(\hat{w}_i\sigma_i \leq 0) \geq \frac{1}{2}$, and for this setting of $\sigma$

$$\mathbb{E}L(\hat{w}) - L^* \geq \frac{1}{4n} \cdot n \cdot \frac{1}{2} \cdot \frac{HB^2}{2n} = \frac{HB^2}{16n} \tag{110}$$

This proves the first term of the first lower bound under Assumptions 1 and 2. For Assumptions 1 and 2, we have instead

$$\mathbb{E}L(\hat{w}) - L^* \geq \frac{HB^2}{16n} = \frac{\Delta}{4} \tag{111}$$

Of course, this latter bound holds only when $n \leq \frac{H}{2\lambda}$.

We note that since $L^* = 0$ in this example, the variance of gradients at the optimum, $\mathbb{E}\|\nabla\ell(w^*; (x, y))\|^2 = 0$. Therefore, these lower bounds hold when the bound on $L^*$ is replaced by the bound $\mathbb{E}\|\nabla\ell(w^*; (x, y))\|^2 \leq \sigma_*^2$.

**The second terms of each bound**    These lower bounds are also both based on the following simple 1-dimensional least squares problem. The loss is given by

$$\ell(w; (x; y)) = \frac{1}{2}(wx - y)^2 \tag{112}$$

The distribution is defined using a sign $\sigma \in \{\pm 1\}$ to be chosen later. With probability $1 - p$, $(x, y) = (0, 0)$, and with probability $p$, $x = \sqrt{H}$ and $y \sim \mathcal{N}(\sigma\sqrt{H}B, s^2)$.

Because $x^2 \leq H$, it is easy to confirm that $\ell(w; (x; y))$ is $H$-smooth, convex, and non-negative, so it satisfies Assumption 1. Also, the expected loss is

$$L(w) = \mathbb{E}_{x,y}\frac{1}{2}(wx - y)^2 = \frac{p}{2}\left(\sqrt{H}w - \sqrt{H}B\sigma\right)^2 + \frac{ps^2}{2} \tag{113}$$

It is easy to see that $L$ is convex and is minimized at $w^* = B\sigma$, which has L2 norm $B$ and the minimizing value is $L(w^*) = \frac{ps^2}{2}$. Therefore, choosing $s^2 = \frac{2L^*}{p}$ ensures that $L$ satisfies Assumption 2.

Alternatively, $L(0) - L^* = \frac{pHB^2}{2}$, so choosing $pB^2 \leq \frac{2\Delta}{H}$ ensures $L(0) - L^* \leq \Delta$. Furthermore,

$$L(w) - L^* = \frac{p}{2}\left(\sqrt{H}w - \sqrt{H}B\sigma\right)^2 = \frac{Hp}{2}\|w - w^*\|^2 \tag{114}$$

Therefore, choosing $p = \frac{\lambda}{H}$ ensures that $L$ is $\lambda$-strongly convex, so it satisfies Assumption 3.

Under either set of assumptions, minimizing $L$ using $n$ samples $(x_1, y_1), \ldots, (x_n, y_n)$ amounts to a Gaussian mean estimation problem using just the subset of $k$ samples for which $x \neq 0$. By Lemma 4, this means that for any algorithm, for some setting of $\sigma \in \{\pm 1\}$,

$$\mathbb{E}[L(\hat{w}) - L^* \mid k] \geq \frac{pHB^2}{2}\left(1 - \sqrt{\frac{HB^2k}{s^2}}\right) \tag{115}$$

Applying Jensen's inequality to the convex function $-\sqrt{k}$, we conclude that

$$\mathbb{E}L(\hat{w}) - L^* \geq \frac{pHB^2}{2}\left(1 - \sqrt{\frac{HB^2np}{s^2}}\right) = \frac{pHB^2}{2}\left(1 - \sqrt{\frac{HB^2np^2}{2L^*}}\right) \tag{116}$$

For Assumptions 1 and 2, we set the remaining parameter as $p^2 = \frac{L^*}{2HB^2n}$ and conclude

$$\mathbb{E}L(\hat{w}) - L^* \geq \sqrt{\frac{HB^2L^*}{32n}} \tag{117}$$

For Assumptions 1 and 3, we consider two cases: If $\Delta \leq \frac{HL^*}{4\lambda n}$, we set $B^2 = \frac{2\Delta}{pH}$ to conclude

$$\mathbb{E}L(\hat{w}) - L^* \geq \Delta\left(1 - \sqrt{\frac{\Delta\lambda n}{HL^*}}\right) \geq \frac{\Delta}{2} \tag{118}$$

Otherwise, we set $B^2 = \frac{L^*}{2\lambda np} \leq \frac{2\Delta}{Hp}$ and conclude

$$\mathbb{E}L(\hat{w}) - L^* \geq \frac{HL^*}{8\lambda n} \tag{119}$$

Therefore, under Assumptions 1 and 2, the loss is at least

$$\mathbb{E}L(\hat{w}) - L^* \geq c \cdot \min\left\{\frac{HL^*}{\lambda n}, \Delta\right\} \tag{120}$$

We note that by Lemma 3, the variance of gradients at the optimum, $\mathbb{E}\|\nabla\ell(w^*; (x,y))\|^2 \leq 2HL^*$. Therefore, when the bound on $L^*$ is replaced by the bound $\mathbb{E}\|\nabla\ell(w^*; (x,y))\|^2 \leq \sigma_*^2$, we have the lower bounds

$$\mathbb{E}L(\hat{w}) - L^* \geq c \cdot \frac{\sigma_* B}{\sqrt{n}} \tag{121}$$

in the convex case and

$$\mathbb{E}L(\hat{w}) - L^* \geq c \cdot \min\left\{\frac{\sigma_*^2}{\lambda n}, \Delta\right\} \tag{122}$$

in the strongly convex case. This completes the proof. $\qquad\square$

# E   Stochastic Optimization with Bounded Variance at the Optimum

So far, we have considered optimizing objectives where the instantaneous losses are non-negative and the value of the minimum of the expected loss is bounded and small, but in other contexts we may want to understand the complexity of optimization in terms of bounds on the variance of the stochastic gradients. In the optimization literature, it is common to assume that the variance of the stochastic gradients is bounded uniformly on the entire space, i.e. $\sup_w \mathbb{E}\|\nabla\ell(w; z)\|^2 \leq \sigma^2$. When, in addition to this variance bound, the objective $L$ is $H$-smooth and convex, and has a minimizer with norm at most $B$, then it has long been known that $T$ steps of SGD achieves error [25]

$$\mathbb{E}L(w_T) - L^* \leq \frac{HB^2}{T} + \frac{\sigma B}{\sqrt{T}} \tag{123}$$

However, the assumption of uniformly upper bounded variance can be strong, and it turns out that when $\ell$ is also $H$-smooth, the $\sigma$ in SGD's guarantee can easily be replaced with $\sigma_*$, an upper bound on the standard deviation of the variance just at the minimizer specifically, i.e. $\mathbb{E}\|\nabla\ell(w^*; z)\|^2 \leq \sigma_*^2$ [8, 14, 21, 23, 31, 34], i.e. SGD guarantees

$$\mathbb{E}L(w_T) - L^* \leq \frac{HB^2}{T} + \frac{\sigma_* B}{\sqrt{T}} \tag{124}$$

Indeed, for other *non-accelerated* algorithms, the weaker bound $\sigma_*$ often suffices and a global variance bound is unnecessary [e.g. 16, 37]. However, it was not clear whether it is possible to make this substitution of $\sigma_*$ for $\sigma$ for *accelerated* methods. For example, Lan [17]'s optimal stochastic first-order algorithm guarantees

$$\mathbb{E}L(w_T) - L^* \leq \frac{HB^2}{T^2} + \frac{\sigma B}{\sqrt{T}} \tag{125}$$

Can we replace this $\sigma$ with $\sigma_*$ too? This would represent a significant improvement. As discussed previously, we can expect $\sigma_*$ to be small—potentially even zero, and anyways often much smaller than $\sigma$—for problems of interest, including training machine learning models in the (near-) interpolation regime. More generally, it is often desirable, and generally much easier, to control the stochastic gradient variance at a single point versus globally. As an example, for "heterogeneous" distributed optimization—where different parallel workers have access to samples from different data

distributions—it is common to bound a measure of the "disagreement" between these different data distributions specifically at the minimizer, which amounts to bounding the variance of the stochastic gradients at $w^*$ [see, e.g., the discussion in 37].

Unfortunately, a consequence of our lower bound, Theorem 4, is that the $\sigma$ in the accelerated rate (125) *cannot* generally be replaced by $\sigma_*$ in the same way as it can be for the unaccelerated rate (124). In fact, since Theorem 4 applies to *any* learning rule that uses $n$ samples, this holds also for non-first-order methods too:

**Corollary 2.** *For $\ell(w; (x, y)) = \frac{1}{2}(\langle w, x \rangle - y)^2$ the square loss, for any learning algorithm that uses $n$ i.i.d. samples, there exists a distribution over $(x, y)$ such that $L$ has a minimizer with norm less than $B$, $\ell$ is $H$-smooth and convex, and $\mathbb{E}\|\ell(w^*; z)\|^2 \leq \sigma_*^2$, and for a universal constant $c$, the algorithm's output has error at least*

$$\mathbb{E}L(\hat{w}) - L^* \geq c \cdot \left( \frac{HB^2}{n} + \frac{\sigma_* B}{\sqrt{n}} \right)$$

*There is also a distribution over $(x, y)$ such that $L$ is satisfies $L(0) - L^* \leq \Delta$, $L$ is $\lambda$-strongly convex, and $\ell$ is $H$-smooth and convex, and for a universal constant $c$, the algorithm's output has error at least*

$$\mathbb{E}L(\hat{w}) - L^* \geq c \cdot \left( \Delta \cdot \mathbb{1}_{n \leq \frac{H}{2\lambda}} + \min\left\{ \frac{\sigma_*^2}{\lambda n}, \Delta \right\} \right)$$

As in Section 5, since a single stochastic gradient estimate $\nabla \ell(w; z)$ can be computed with one sample, this lower bound also applies to minibatch first-order algorithms with $n = bT$, and the lower bound (10) for deterministic first-order optimization still holds so we also have

**Corollary 3.** *For any algorithm that uses $T$ minibatch stochastic gradients of size $b$, there exists an objective $L(w) = \mathbb{E}_z \ell(w; z)$ where $L$ has a minimizer with norm less than $B$, $\ell$ is $H$-smooth and convex, and $\mathbb{E}\|\ell(w^*; z)\|^2 \leq \sigma_*^2$, so that for a universal constant $c$, the algorithm's output has error at least*

$$\mathbb{E}L(\hat{w}) - L^* \geq c \cdot \left( \frac{HB^2}{T^2} + \frac{HB^2}{bT} + \frac{\sigma_* B}{\sqrt{bT}} \right)$$

*There is also an objective that satisfies $L(0) - L^* \leq \Delta$, $L$ is $\lambda$-strongly convex, and $\ell$ is $H$-smooth and convex, so that for universal constants $c, c'$, the algorithm's output has error at least*

$$\mathbb{E}L(\hat{w}) - L^* \geq c \cdot \left( \Delta \exp\left( -\frac{c'\sqrt{\lambda}T}{\sqrt{H}} \right) + \Delta \cdot \mathbb{1}_{bT \leq \frac{H}{2\lambda}} + \min\left\{ \frac{\sigma_*^2}{\lambda bT}, \Delta \right\} \right)$$

Ignoring again the small gap between $\exp(-\frac{c'\lambda bT}{H})$ and $\mathbb{1}_{bT \leq \frac{H}{2\lambda}}$ (see the discussion below Corollary 1), this shows, in essence, that when $b = 1$, it is impossible to achieve the accelerated optimization rates of $T^{-2}$ and $\exp(-\sqrt{\lambda}T/\sqrt{H})$ under the conditions of Corollary 3. Furthermore, when $b = 1$, the guarantee of regular, unaccelerated SGD actually matches the lower bound, so there is no room for acceleration, Lan's accelerated SGD algorithm relied crucially on the uniformly bounded variance, and the $\sigma$ in (125) cannot generally be replaced with $\sigma_*$. In fact, Corollary 2 shows that no learning rule, even non-first-order methods, can ensure error $n^{-2}$ using just $n$ samples.

However, the good news is that our guarantees for *minibatch* accelerated SGD also apply in this setting:

**Theorem 5.** *Let $L(w) = \mathbb{E}_z \ell(w; z)$ have a minimizer with norm at most $B$, let $\ell$ be $H$-smooth and convex, and let $\mathbb{E}\|\nabla \ell(w^*; z)\|^2 \leq \sigma_*^2$. Then Algorithm 1 guarantees*

$$\mathbb{E}L(w_T^{ag}) - L^* \leq c \cdot \left( \frac{HB^2}{T^2} + \frac{HB^2}{bT} + \frac{\sigma_* B}{\sqrt{bT}} \right)$$

*Let $L(w) = \mathbb{E}_z \ell(w; z)$ satisfy $L(w) - L^* \geq \frac{\lambda}{2} \min_{w^* \in \arg\min_w L(w)} \|w - w^*\|^2$ for all $w$, let $L(0) - L^* \leq \Delta$, let $\ell$ be $H$-smooth and convex, and let $\mathbb{E}\|\nabla \ell(w^*; z)\|^2 \leq \sigma_*^2$. Then $\mathsf{GC2Cvx}(Alg\,1, e)$ guarantees*

$$\mathbb{E}L(\hat{w}) - L^* \leq c \cdot \left( \Delta \exp\left( -\frac{c'\sqrt{\lambda}T}{\sqrt{H}} \right) + \Delta \exp\left( -\frac{c'\lambda bT}{H} \right) + \frac{\sigma_*^2}{\lambda bT} \right)$$

The first part of the Theorem is demonstrated in the proof of Theorem 1 in Appendix A, and the second part follows an essentially identical argument as in the proof of Theorem 3.

Corollary 3 showed that it is impossible to achieve error like $T^{-2}$ using first-order methods with $b = 1$. However, Theorem 5 shows it *is* possible to achieve error like $T^{-2}$ with parallel runtime $T$ using our minibatch accelerated SGD method with $b > 1$. In other words, while SGD with minibatches of size $b = 1$ matches the lower bound in Corollary 2 with $n = bT = T$, and therefore attains the smallest possible error using $n$ samples, our method is able to more quickly attain this same optimal error using $n = bT$ samples with $b \gg 1$. As discussed in Section 6, this means our algorithm's parallel runtime, $T$, can be much smaller than SGD's, with up to a quadratic improvement. Since the lower bound, Corollary 3, and upper bound, Thoerem 5, match, this also tightly bounds the complexity of stochastic first-order optimization with a bound on $\sigma_*$.