# OpenReview forum: "An Even More Optimal Stochastic Optimization Algorithm: Minibatching and Interpolation Learning"
_NeurIPS.cc/2021/Conference — NeurIPS 2021 Poster_

### Official Review · Reviewer_1bhL · 2021-07-12

**Rating:** 6
**Confidence:** 4

**Summary:**

This paper studies the behavior of accelerated minibatch SGD under smoothness, bounded variance of gradient at optimum and/or quadratic growth assumptions. They also provide a meta algorithm to generate fast methods of solving convex optimization problems satisfying growth conditions from methods of solving convex problems without growth conditions.

**Limitations And Societal Impact:**

See main review for weaknesses.

**Main Review:**

This paper is technically sound and clearly written.

Strengths:
1. Analysis of minibatch accelerated gradient methods under a slightly different set of assumptions with justification using examples for those assumptions.
2. A general technique to create algorithms for problems satisfying growth conditions using known algorithms to minimize convex functions.
3. They also provide lower bounds showing optimality of convergence rates.

Weaknesses: (in decreasing order of importance)
1. Although the assumption of bounded variance of the gradient only at optimum is weaker than having bounded variance of the gradient at all points, the assumption of having each sample function being smooth is stronger than having only the population function being smooth as is in some of the referred papers. It is unclear to me why one set of assumptions is better than the other. The only example explaining the assumption is that of least squares regression, but having $||w^*|| \leq B $, we can upper bound the dependence of the gradient on $||w||$ by considering a bounded domain. More examples showing the benefits of this set of assumptions compared to the previous ones would be useful.
2. It is not very clear to understand what is the one line message of the paper. It seems like having 3 seemingly disjoint contributions, especially section 7, which seems a bit disconnected from the rest of the paper. Some effort can be made to make the paper gel together better.
3. Interpolation problems are only defined in the special case when $\mathcal{D}$ is the empirical distribution and not for the general case. Having interpolation problems in the title is slightly misleading because this can confuse the reader regarding whether the results hold only for ERM or for population loss minimization. Either having a general \textit{definition} of interpolation problems and checking whether the results go through for that or removing the stress on interpolation problems would be useful.
3. Some of the results are not compared to related work on methods for interpolation problems with growth conditions listed below:

[1] D. Davis and D. Drusvyatskiy. Stochastic model-based minimization of weakly convex
functions. SIAM Journal on Optimization, 29(1):207–239, 2019.
[2] H. Asi and J. C. Duchi, Stochastic (approximate) proximal point methods: Convergence, optimality, and adaptivity, SIAM Journal on Optimization, 29 (2019), pp. 2257–2290, https://arXiv.org/abs/1810.05633.
[3] Karan Chadha, Gary Cheng, and John C Duchi. Accelerated, optimal, and parallel: Some results on model- based stochastic optimization. arXiv preprint arXiv:2101.02696, 2021.
[4] S. Vaswani, F. Bach, M. Schmidt. Fast and Faster Convergence of SGD for Over-Parameterized Models (and an Accelerated Percepton).
[5] S. Vaswani, A. Mishkin, I. Laradji, M. Schmidt, G. Gidel, S. LaCoste-Julien. Painless Stochastic Gradient: Interpolation, Line-Search, and Convergence Rates.


**Time Spent Reviewing:**

2

---

> ### Author Response · Authors · 2021-08-09
> **Author response to reviewer 1bhL**
>
> Thank you for your comments!
>
> 1:
>
> The requirement that the components are smooth (and not only the objective):
>
> As we discuss around line 81, this is a natural assumption in machine learning and holds well beyond least squares, whenever we are training a model using a smooth loss function (e.g. logistic loss, Huber loss, smoothed hinge loss, etc.).
>
> As you say, in many situations, including the one considered by [Lan 2012], it is enough to require the objective is smooth, and so Lan’s results cannot be obtained as a corollary of ours (and we did not claim this). By the same token, Lan’s analysis has no notion of a “component”, $\ell$, and relies only on unbiased estimates of the gradient.  Since we want to get L*-sensitive methods and analysis (i.e. optimistic bounds, bounds for realizable or near-realizable learning, or for interpolation learning), we must discuss individual components, since for the L* bound to be meaningful, we must require individual components to be non-negative. In addition, in order to use L* to control the variance of  the gradients, it seems unavoidable to assume that the components are smooth, and all of the work on optimistic/realizable rates that we are familiar with requires this, see e.g., [Srebro, Sridharan, & Tewari 2010], [Cotter et al. 2011], [Liu & Belkin 2018]).  Since, as mentioned above, this is the natural and typical case in machine learning, we feel this is an interesting setting to consider (individual components smooth and non-negative + L* small), regardless of whether it subsumes other settings which are not L*-dependent.  (see discussion around line 92 and our answer to Reviewer vQ2S regarding the importance of L* dependent methods and analysis in machine learning).
>
> Regarding bounding the variance by relying on ||w*||<=B:
>
> As we discuss around line 126, a bound ||w*||<=B plus smoothness of \ell can indeed be used to bound the variance of the gradient on a ball with radius O(B). However, this bound on the variance will generally be quite large, and plugging such a bound into Lan’s existing analysis for the uniformly bounded variance setting yields suboptimal results, and is thus not satisfying and does not recover the optimal rate.
>
> 2:
>
> The one-line message of the paper is stated in the abstract: we present an algorithm for minibatch stochastic optimization, that is L*-dependent (as is appropriate in many machine learning contexts, and in interpolation learning), it strictly improves over previous methods in the same setting, it is provably optimal, and it yields improved parallelization speedups.
>
> Section 7 is indeed a “bonus” section presenting an extension to a more general setting, which might be of interest to researchers coming at the problem from a different perspective.
>
> 3:
>
> As you say, “interpolation learning” as it is commonly discussed is precisely captured by \mathcal{D} being the uniform distribution over the training samples, so L is the empirical loss, with L*=0 (see lines 60-68).  Our setting thus captures empirical loss minimization using stochastic minibatch gradient methods in interpolation learning.
>
> However, this is not the *only* situation captured by our setting and analysis, which applies equally well when \mathcal{D} is the population data distribution. In this case, the assumption that L*=0 (or L* is small) is more often referred to as “realizability” (or “near-realizability”), but our analysis, algorithm, and discussion are just as applicable in this setting too. So, for the purpose of our paper, the distinction between L being the population loss versus the training loss is just a matter of terminology (“interpolation learning” vs “realizable learning”), and our results hold equally well for either.
>
> In the abstract we say that our results can be “*applied* to interpolation learning” (rather than only being relevant there), and with the recent interest in optimizing the training error in the interpolation setting (e.g. [Liu and Belkin 2018]), we do indeed want to emphasize this application, which for us was a major motivator for this work.
>
> 4:
>
> Thank you for pointing out these references. We will add comments/citations to them in the updated version of the paper where they are relevant.

---

### Official Review · Reviewer_GtzK · 2021-07-17

**Rating:** 7
**Confidence:** 4

**Summary:**

This paper studies an accelerated stochastic gradient method with minibatching for expected risk minimization problems and provides the optimal convergence rate. Remarkable points are that (i) the dependency of the obtained rate on the minibatch size is also optimal, meaning the linear speedup by increase the minibatch size, and (ii) the rate reflects the acceleration due to the interpolation by evaluating the variance at the solution using the optimal value of the objective.

**Limitations And Societal Impact:**

The authors adequately addressed the limitations and potential negative social impact.

**Main Review:**

[Contributions]

Since the minibatching technique is frequently adopted in machine learning applications, the research into minibatch efficiency is important. In addition, relaxation of the uniform boundedness assumption on the noise is also an interesting topic. In this context, I think this study makes a good contribution. Concretely, the method achieves the optimality regarding the number of iterations $T$ and the minibatch size $b$ and shows the acceleration due to the interpolation by taking into account the optimal value of the objective.

Accelerated stochastic gradient methods for the expected risk minimization were studied in Lan [14] and Cotter et al. [8]. Compared to these studies, the obtained rate has advantages as follows:

- The rate in Lan [14] is optimal in $T$ but is suboptimal in $b$. Moreover, there is no acceleration due to the interpolation because [14] uses the uniform boundedness on the noise instead of the evaluation of the noise at the optimal solution.
- The rate in Cotter et al. [8] is optimal in $T$ and accelerated by the interpolation, but is suboptimal in $b$.

Hence, I think this submission makes a certain contribution in the context.

[Additional references]

There are several missing references.

For finite-sum problems, some methods achieve the accelerated convergence rate and linear speed up by the minibatching simultaneously. For instance, see the following papers:

- Z. Allen-Zhu. Katyusha: The first direct acceleration of stochastic gradient methods, 2017.
- T. Murata and T. Suzuki. Doubly accelerated stochastic variance reduced dual averaging method for regularized empirical risk minimization, 2017.
- A. Nitanda, T. Murata, T. Suzuki. Sharp Characterization of Optimal Minibatch Size for Stochastic Finite Sum Convex Optimization, 2019.

The minimum minibatch size to achieve the optimal iteration complexity was discussed in the last paper.

The linear speedup by increasing minibatch size up to a certain size was shown for SAGA in the following paper, although they only consider the non-accelerated method.

- N. Gazagnadou, R. M. Gower, and J. Salmon. Optimal Mini-Batch and Step Sizes for SAGA, 2019.

Acceleration due to the interpolation was also considered in the relaxed condition, i.e., PL-condition in the following:

- R. Bassily, M. Belkin, S. Ma. On exponential convergence of SGD in non-convex
over-parametrized learning, 2018.

The faster convergence under strong and weak growth conditions was shown in

- S. Vaswani, F. Bach, and M. Schmidt. Fast and Faster Convergence of SGD for Over-Parameterized Models (and an Accelerated Perceptron), 2018.

These conditions are closely related to the interpolation.

Moreover, an extension of the weak growth condition, which does not require perfect interpolation was also considered in

- R.M. Gower, N. Loizou, X. Qian, A. Sailanbayev, E. Shulgin, and P. Richtarik. SGD: General Analysis and Improved Rates, 2019.

**Time Spent Reviewing:**

5

---

> ### Author Response · Authors · 2021-08-09
> **Author response to reviewer GtzK**
>
> Thank you for your comments and the list of additional references! We will add some comments and references to these related papers where they are relevant. Some comments:
>
> Unless we are missing something, the finite sum papers ([Allen-Zhu], [Murata & Suzuki], [Nitanda et al.], and [Gazagnadou et al.] do not seem to address the realizable/near-realizable setting (i.e. with bounded L*), but the role of the minibatch size does seem related in some ways. We will comment on these in Section 6.
>
> The [Bassily et al.] paper considers a similar setting as ours, but with the PL condition instead of our Assumption 3. However, their methods are not “accelerated” in the sense of depending on the square root of the condition number (or the analogue of the condition number for PL objectives: smoothness / PL_constant). We will mention this in the related work section.
>
> The [Vaswani et al.] and [Gower et al.] papers are definitely related and we will also discuss them in the related work section. This may already be obvious to you, but it seems that their notion of “growth conditions'' are different than ours, and it is also not immediately clear how the Vaswani papers results would extend to the case of minibatching given their particular assumptions about the noise in the stochastic gradients.

---

> > ### Comment · Reviewer_GtzK · 2021-08-27
> > **Thanks for the response.**
> >
> > Thanks for the response.
> > These comments on related studies are useful and make the position of the paper clearer. I hope that they will be incorporated in the revised version. I don't think there are any comments in the other reviews which hurt the contribution of the paper.
> > I would like to keep the score.

---

### Official Review · Reviewer_vQ2S · 2021-07-17

**Rating:** 6
**Confidence:** 3

**Summary:**

This paper improves the complexity bound w.r.t. minibatch speedup given by Cotter et al and Liu et al. for minibatch accelerated gradient descent for certain objectives (under sharpness condition). Generally the paper is well-organized with solid theoretical analysis and is easy to follow.



**Limitations And Societal Impact:**

Perhaps the application of this approach does not include regularized problems in which the objective is far bigger than zero. For example, underdertermined linear system is mentioned in the paper. In practice we can add $\ell_1$ term to enforce sparsity, then we have $L^*>>0$.

**Main Review:**

The structure of this paper is very clear.  It presents sharper complexities of accelerated sgd than state-of-the-art result by using a refined bound on stochastic noice. The technical analysis is solid, from the theoretical aspect, I think the improvement is quite interesting and the contribution is sufficient. However, it would be much better if the author can empirically verify the algorithm performance.

The paper tries to design the minibatch algorithm with fixed batch size, can the author also briefly discuss how to use varying batch size, this seems to be a more practical strategy in applications.

I am a little concerned about the parameters in this paper and the assumptions. For example, in assumption 2, it requires some nonstandard parameters such as L^* and B (Assumption 2), which are usually unknown in practice.  In assumption 3, the algorithm also needs the growth condition.

I understand that the paper aims to obtain the sharpest bound possible, but it would also be great if it also discuss what if those parameters are under/overestimated. Would that has a significant impact on the performance?

Can the author also provide some preliminary experiments to to show the superior performance of the proposed approach.


On eq (4), $\ell(w,z)\Rightarrow \nabla\ell(w,z)$.

Eq (59) $\nabla F(w_t^{md})\Rightarrow \nabla L(w_t^{md})$.

Line 300 $E\|\nabla \ell(w^*;z)\|^2$


**Time Spent Reviewing:**

6

---

> ### Author Response · Authors · 2021-08-09
> **Author response to reviewer vQ2S**
>
> Thank you for your comments and for pointing out the typos, which we will fix.
>
> Regarding experiments, please see bullet #2 in our response to reviewer re6H.
>
> Regarding varying the minibatch size:
>
> This is a very interesting but somewhat difficult question to answer. In the more standard stochastic optimization literature, where there is a uniform upper bound on the variance of sigma^2, minibatching is exactly equivalent to reducing the variance of the gradient estimates from sigma^2 -> sigma^2 / b. In our case, the effect of minibatching is more complicated, and it is non-trivial to see how changing the minibatch size over time would affect the algorithm’s performance. It seems quite possible that using larger minibatches at the beginning (when the gradient variance will be larger) and then using smaller minibatches later on (when the gradient variance is smaller) might be advantageous, but it will require a non-trivial extension to our analysis to figure this out. We will think more about this, thanks for raising the point!
>
> Regarding our assumptions:
>
> By B and L* being “non-standard”, do you mean that it is not standard to base our theoretical analysis on them and to look at optimizing classes of problems defined in terms of B and L*, or rather that it is “non-standard” to know these values, and for the method hyperparameters to be based on them?
>
> Regarding basing the analysis on B and L* (perhaps this is already obvious and not the issue you are concerned with):
>
> The combination of convexity with the bound, B, is extremely standard in the convex optimization literature. It is used in the papers that we most closely compare to: [Lan 2012] and [Cotter et al. 2011]; it is used in many textbooks, e.g., [Nesterov 2004], [Bubeck 2014]; and it is used in innumerable papers, including, e.g., a standard citation for GD and SGD analysis [Nemirovsky & Yudin 1983]. As discussed around lines 107-119, Assumption 3 is strictly weaker than the (also very standard) strong convexity assumption, but is actually sufficient for obtaining similar rates, and better suited for learning problems such as underdetermined least squares, and so is arguably a better assumption.
>
> The realizability (or near-realizability) assumption, L*=0 (or L* small), is less standard in the optimization literature, but, this stands at the heart of the paper, and we are promoting it’s study from an optimization perspective (although we are by no means the first to consider it in this optimization context). This is a natural assumption in ML, where we are interested in training models that can actually capture the data well, and goes all the way back to the original work on PAC learning [Valiant 1984]. There has been significant work and interest in ML is showing faster rates under realizability (L*=0) or near-realizability (low L*) (see discussion on lines 20-33 and citations therein).  Thinking of the objective function L(w) as the empirical loss (training error), the assumptions has gained renewed interest with the interest from the perspective of interpolation learning (e.g. [Liu and Belkin 2018]), and noting that many modern ML approaches we train to within zero, or very small, training error. It is thus important to understand how minimizing the empirical error is easier under the assumption that it can actually be brought down to zero, or close to zero.
>
> Regarding knowing the parameters:
>
> This is indeed more of a concern, but unfortunately many methods, including [Cotter et al. 2011], [Lan 2012], and ours, rely on knowing such parameters.  For a discussion of this, and whether it might be avoidable, see bullet #1 in our response to reviewer re6H.
>
> Regarding L* when regularization is used:
>
> This is an interesting point. The way that we used L* in our analysis was to control the noise in the stochastic gradient estimates (via Lemma 3). Because an added regularizer is “known” and exact/noiseless gradients of the regularizer are available, we suspect that all of our analysis could be extended to show that the contribution of the regularizer to the minimum value of the loss can be ignored. More specifically, we suspect that for objectives of the form F(w) = L(w) + R(w), where L is as defined in our paper, R is convex and H smooth, and exact gradients of R are available to the algorithm, then our guarantees should hold (perhaps up to constant factors) in terms of L* = min_w L(w), i.e. regardless of the value of R at the minimum.

---

### Official Review · Reviewer_re6H · 2021-07-20

**Rating:** 7
**Confidence:** 4

**Summary:**

This paper presents and analyzes an accelerated algorithm for convex and strongly convex minimization when the objective function is an expectation of a convex function. It claims that the algorithm is optimal w.r.t. Mini-batch size and the convergence rate. The presented algorithm reaches the best-known rate for the accelerated stochastic algorithm and also shows the dependence on the mini-batch size is also optimal.

**Limitations And Societal Impact:**

Yes

**Main Review:**

Significance: The paper considers minimizing the population risk in machine learning models, which is one of the major optimization problems in ML.

Novelty: The paper presents a novel algorithm and also discovers a new upper bound for the second moment of the stochastic gradient. Using this upper bound it analyzes the presented algorithm under different assumptions and recovers the optimal upper bounds.

Clearity: it is easy to follow and understand all the technical details and main text of the paper. However, there are some typos here and there that need to be fixed.




My question

1- The method has a limitation in the sense that it requires knowing L*. Could you replace this with some upper or lower bounds?

2- There is no experimental section. It would be helpful to see the performance of the algorithm on the model with real data.

3- Your algorithm is not an anytime algorithm. How can you make it an anytime algorithm?

4-  It is not clear why theorem 2 is not trivial based on the alg 2.




**Time Spent Reviewing:**

5

---

> ### Author Response · Authors · 2021-08-09
> **Author response to reviewer re6H**
>
> Thank you for your comments and interesting questions.
>
> 1- Indeed, the optimal stepsize depends on $L^*$ (and $H$ and $B$), which may not be known a priori. Our method can still be used when $L^*$, $H$, and/or $B$ are unknown, with a corresponding degradation in the convergence rate. Our analysis holds for any upper bound $\tilde{L} \geq L^*$ and $\tilde{B} \geq B$, with $\tilde{L}$ and $\tilde{B}$ replacing $L^*$ and $B$ in the method and its convergence rate. Also, Equation 77 in the appendix (with ${\sigma^*}^2$ <-> $2HL^*$) shows an upper bound on the error (in terms of the actual $L^*$ and $B$, even when they are unknown) for any choice of the parameter $\gamma \leq \min${ $1/(12H), b/(24H(T+1))$ }. So, all that is really needed to use our method is an upper bound on $H$ (which you would need for any other first-order method that we can think of), but the guarantee will be worse if only a poor estimate of $L^*$ or $B$ are available. We will add a discussion of this issue below Theorem 1 in the updated version.
>
> We would like to point out that this issue is not unique to our algorithm, and many existing accelerated stochastic optimization algorithms also require knowing these in order to get optimal results. For example, for [Lan 2012]’s  AC-SA algorithm to attain the optimal rate for the typical uniformly bounded variance setting (see discussion around line 122), the stepsizes need to be chosen using knowledge of H, B, and sigma^2 (the uniform bound on the gradient variance, which is analogous to L* in our setting). Likewise, [Cotter et al. 2011]’s algorithm and its guarantee (see discussion around line 130) require setting the stepsize based on L*, H, and B.
>
> Finally, our stepsize/momentum parameters are chosen carefully based on the problem parameters in order to make the analysis work, however, people typically don’t use the theoretically-prescribed stepsizes/momentum parameters in practice (e.g. when using SGD, SGD w/ momentum, Adam, etc), and it is much more common to tune them using validation data. In experiments (see next bullet), we (unsurprisingly) obtained the best results when we tuned the parameters rather than using the theoretical ones, so we don’t think the lack of knowledge of L* or B would be a huge problem practically.
>
> 2- We conducted some numerical simulations (synthetic least-squares and binary logistic regression problems) and our results are extremely similar to the related work of [Cotter et al. 2011], which we discuss in our draft. With the theoretically-prescribed stepsize/momentum parameters, there is an advantage for our method over Cotter et al.’s.  But, both our method and Cotter’s methods perform better when these parameters are tuned, and in this case the methods are the same, since we differ from Cotter et al only in stepsize/momentum schedules and the analysis. Hence, our main contribution and focus in this paper was getting tighter theoretical guarantees for our algorithm, getting matching upper and lower bounds on the minimax error, and seeing how these rates depend on the various problem parameters that we consider.  On the practical side, having tighter and better analysis gives better guidance for selecting the parameters, but it is very difficult to have a fair and objective quantitative comparison of “better guidance for selecting parameters”.
>
> 3- Indeed, our theoretical guarantee requires knowledge of the time horizon (as do many other accelerated stochastic algorithms like [Lan 2012] and [Cotter et al. 2011]). The classic “doubling trick” can be used to convert our method to an anytime algorithm: run with horizon T=1 for one iteration, then start over with horizon T=2 for two iterations, and so on starting over with horizon T=2^k and running for 2^k iterations. While this is a little awkward, it does allow you to get our same guarantee (up to a small constant factor) for any time. A more direct anytime method would be nice to have, but this would require a non-trivial modification to the analysis, and we are not aware of other accelerated stochastic first-order methods that have anytime guarantees.
>
> That said, in our simulations (see bullet #2 and the last paragraph of bullet #1 above) our algorithm works best using tuned stepsize/momentum parameters, so not knowing T in advance may not be a huge problem practically.
>
> 4- You’re right that it is a simple argument, we included a proof for completeness’s sake.

---

### Decision · Program_Chairs · 2021-09-27

**Decision:**

Accept (Poster)

**Comment:**

All reviewers recommend accepting the paper. Please take the time to consider the reviewer's comments and update the paper when preparing the final version. In particular, please discuss the additional related references brought up in the reviews.